# Evaluating the simulated radiative forcings, aerosol properties and stratospheric warmings from the 1963 Mt Agung, 1982 El Chichón and 1991 Mt Pinatubo volcanic aerosol clouds

Sandip S. Dhomse[1,2], Graham W. Mann[1,3], Juan Carlos Antuña Marrero[4], Sarah E. Shallcross[1], Martyn P. Chipperfield[1,2], Kenneth S. Carslaw[1], Lauren Marshall[1,5], N. Luke Abraham[5,6], and Colin E. Johnson[3,7]

[1]School of Earth and Environment, University of Leeds, Leeds, UK
[2]National Centre for Earth Observation, University of Leeds, Leeds, UK
[3]National Centre for Atmospheric Science (NCAS-Climate), University of Leeds, UK
[4]Department of Theoretical Physics, Atomic and Optics, University of Valladolid, Valladolid, Spain
[5]Department of Chemistry, University of Cambridge, Cambridge, UK
[6]National Centre for Atmospheric Science, University of Cambridge, UK
[7]Met Office Hadley Centre, Exeter, UK

**Correspondence:** Sandip Dhomse (s.s.dhomse@leeds.ac.uk), Graham Mann (g.w.mann@leeds.ac.uk)

**Abstract.** Accurately quantifying volcanic impacts on climate is a key requirement for robust attribution of anthropogenic climate change. Here we use the UM-UKCA composition-climate model to simulate the global dispersion of the volcanic aerosol clouds from the three largest eruptions of the 20th century: 1963 Mt. Agung, 1982 El Chichón and 1991 Mt. Pinatubo. The model has interactive stratospheric chemistry and aerosol microphysics, with coupled aerosol-radiation interactions for realistic composition-dynamics feedbacks. Our simulations align with the design of the Interactive Stratospheric Aerosol Model Intercomparison (ISA-MIP) "Historical Eruption $SO_2$ Emissions Assessment". For each eruption, we perform 3-member ensemble model experiments for upper, mid-point and lower estimates of $SO_2$ emission, each initialised to a meteorological state to match the observed phase of the quasi-biennial oscillation (QBO) at the times of the eruptions. We assess how each eruption's emitted $SO_2$ translates into a tropical reservoir of volcanic aerosol and analyse the subsequent dispersion to mid-latitudes.

We compare the simulations to different volcanic forcing datasets (e.g. Space-based Stratospheric Aerosol Climatology (GloSSAC), Sato et al. (1993) and Ammann et al. (2003)) that are used in historical integrations for the two recent Coupled Model Intercomparison Project (CMIP) assessments. We assess the vertical extent of the volcanic aerosol clouds by comparing to Stratospheric Aerosol and Gas Experiment II (SAGE II) v7.0 satellite aerosol extinction measurements (1985-1995) for Pinatubo and El Chichón, and to 1964-65 northern hemisphere ground-based lidar measurements for Agung. As an independent test for the simulated volcanic forcing after Pinatubo, we also compare to the shortwave (SW) and longwave (LW) top-of-the-atmosphere flux anomalies measured by the Earth Radiation Budget Experiment (ERBE) satellite instrument.

For the Pinatubo simulations, an injection of 10 to 14 Tg $SO_2$ gives the best match to the High Resolution Infrared Sounder (HIRS) satellite-derived global stratospheric sulphur burden, with good agreement also to SAGE II mid-visible

and near-infrared extinction measurements. This 10-14 Tg range of emission also generates a heating of the tropical stratosphere that is comparable with the temperature anomaly present in the ERA-Interim reanalyses. For El Chichón, the simulations with 5 Tg and 7 Tg $SO_2$ emission give best agreement with the observations. However, for first few months these simulations predict a much deeper volcanic cloud than represented in the GloSSAC dataset that is largely based on an interpolation between Stratospheric Aerosol Mesurements (SAM-II) satellite and aircraft measurements. In contrast, these simulations show much better agreement during the SAGE II period after October 1984. For 1963 Agung, the 9 Tg simulation compares best to the forcing datasets with the model capturing the lidar-observed signature of the altitude of peak extinction descending from 20 km in 1964 to 16 km in 1965.

Overall, our results indicate that the downward adjustment to $SO_2$ emission found to be required by several interactive modelling studies when simulating Pinatubo, is also needed when simulating the Agung and El Chichón aerosol clouds. This strengthens the hypothesis that interactive stratospheric aerosol models may be missing an important removal or redistribution process (e.g. effects of co-emitted ash) which changes how the tropical reservoir of volcanic aerosol evolves in the initial months after an eruption. Our model comparisons also identify potentially important inhomogeneities in the CMIP6 dataset for all three eruption periods that are hard to reconcile with variations predicted in the interactive stratospheric aerosol simulations. We also highlight large differences between the CMIP5 and CMIP6 volcanic aerosol datasets for the Agung and El Chichón periods. Future research should aim to reduce this uncertainty by reconciling the datasets with additional stratospheric aerosol observations.

## 1  Introduction

Quantifying the effects of volcanic eruptions on the climate system is challenging due to complex coupling pathways between various atmospheric processes (Cadle and Grams, 1975; Turco et al., 1982; Robock, 2000). All major volcanic eruptions directly inject large amounts of $SO_2$ into the stratosphere, leading to an abrupt enhancement of the stratospheric aerosol layer. The principal effect of volcanic aerosol clouds is to increase backscatter of incoming solar radiation, thereby cooling the Earth's surface. Major volcanic aerosol clouds can also cause a range of other composition responses, which together with the direct aerosol effects, initiate a complex system of radiative, dynamical and chemical interactions. As aerosol particles in the volcanic cloud grow larger, they also absorb outgoing longwave (LW) radiation, which offsets some of the shortwave (SW) surface cooling, and also causes a warming of the lower stratosphere (e.g. Angell, 1997a; Free and Lanzante, 2009). When this volcanic-aerosol-induced heating occurs within the tropical stratospheric reservoir (e.g. Dyer, 1974; Grant et al., 1996), the effect causes an increase in the upwelling in the lowermost tropical stratosphere. Such tropical stratospheric warmings also alter the meridional temperature gradient in the stratosphere, which in turn can modify the vertical propagation (and breaking) of the large planetary and synoptic-scale waves that drive the Brewer-Dobson circulation (e.g. Poberaj et al., 2011; Bittner et al., 2016), with decreased tropical ozone and additional ozone transport to mid-latitudes caused by the enhanced upwelling (e.g. Kinne et al., 1992; Dhomse et al., 2015). These indirect (circulation-driven) ozone changes combine with direct chemical ozone loss from the increased aerosol surface area

available for heterogeneous chemistry (e.g. Prather, 1992; Solomon, 1999), and also photochemical ozone changes (e.g.
Bekki et al., 1993).

Eruptions that inject $SO_2$ directly into the tropical stratosphere cause relatively prolonged surface cooling, because a long-lived "reservoir" of volcanic aerosol forms (Dyer, 1974; Grant et al., 1996), with particles in the volcanic cloud remaining within the "tropical pipe" due to a prevailing and sustained upwelling (Plumb, 1996). At the edge of the tropical pipe, the "subtropical-barrier" reduces transport to mid-latitudes, slowing subsequent removal via stratosphere-troposphere ex-
60 change (Holton et al., 1995). Since the intensity of incoming solar radiation is highest at low latitudes, a tropical volcanic aerosol cloud also has greatest solar dimming efficacy. The three largest tropical eruptions over the past century are Mt. Agung (March 1963), El Chichón (April 1982) and Mt. Pinatubo (June 1991). The extent to which these eruptions cool the Northern and Southern Hemispheres differ substantially depending to a large extent on the dispersion pathways of the resulting volcanic aerosol clouds from the tropical reservoir. For El Chichón and Agung, the volcanic aerosol dispersed
mostly to the hemisphere of the volcano (e.g. Dyer, 1970; McCormick and Swissler, 1983), whereas for Pinatubo the cloud dispersed to both hemispheres (e.g. Trepte et al., 1993).

Major eruptions are known to cause dominant cooling signatures within decadal global mean surface temperature (GMST) trends (e.g. Santer et al., 2001, 2014). The abruptness and dominant magnitude of major volcanic forcings, compared to the slower variations in all other external forcings, means even a small relative uncertainty in their global dimming
impact will introduce important variations in the decadal GMST tends (e.g. Marotzke and Forster, 2015). There has been a substantial change in the volcanic forcing from 1963 Agung between CMIP5 and CMIP6 (Niemeier et al., 2019), and the effects that this change may have caused within CMIP5 and CMIP6 historical simulations is starting to become recognised (e.g. Mann et al., 2020). Even with the greater amount of observational data after the most recent major eruption (Pinatubo), the magnitude of the peak stratospheric aerosol optical depth (sAOD) remains highly uncertain at 0.25 - 0.45
(e.g. Russell et al., 1996; Kovilakam et al., 2020). Global tropospheric cooling estimates from Pinatubo are even more uncertain, ranging from 0.2 K - 0.5 K (Soden et al., 2002; Canty et al., 2013; Folland et al., 2018). The modern satellite era has provided a wealth of information about the progression of volcanic aerosol clouds, but space-borne remote sensing measurements can sometimes have significant uncertainties. After the 1991 Pinatubo eruption, the unprecedented optical thickness of the volcanic aerosol cloud caused retrieval problems for several limb-sounding satellite instruments.
For example, the Stratospheric Aerosol and Gas Experiment (SAGE-II) instrument that provides the benchmark dataset for Pinatubo, was only able to measure aerosol extinction in the upper parts of the tropical volcanic cloud (e.g. Thomason, 1992). Nadir-sounding satellite measurements such as the Advanced Very High Resolution Radiometer (AVHRR) provide important information for the dispersion of the El Chichón (Robock and Matson, 1983) and Pinatubo (e.g. Long and Stowe, 1994) aerosol clouds, but are not able to determine their vertical distribution.
Another important uncertainty for Pinatubo's effects is the magnitude and longevity of the warming of the lower stratosphere. Within the CCMVal-2 hindcast integrations (SPARC, 2010, Chap. 8), chemistry-climate model show about 0.5 to 3 K warming at 50 hPa, with the temperature anomaly from the ERA-interim reanalysis, (Dee et al., 2011) suggesting ∼1 K warming. The magnitudes of the lower stratospheric warmings for the El Chichón and Agung eruptions are

even more uncertain (e.g. Free and Lanzante, 2009; Driscoll et al., 2012; DallaSanta et al., 2019). The large diversity in the CCMVal-2 warming anomalies are mostly due to differences in the methodologies used to estimate the volcanic heating, differences in vertical resolution and stratospheric circulation, meaning the effects from the quasi-biennial oscillation (QBO) phase propagation (Angell, 1997a; Sukhodolov et al., 2018), and influences from 11-year solar flux variability, (e.g. Lee and Smith, 2003; Dhomse et al., 2011, 2013) are resolved differently. The attribution of volcanically forced warming is also complicated by the fact that the increased tropical upwelling caused by the aerosol-induced heating subsequently leads to changes to the volcanic aerosol cloud itself (e.g. Young et al., 1994; McCormick et al., 1995; Aquila et al., 2013), a partial offset of the warming also caused by a circulation-driven reduction in tropical ozone (e.g. Kinne et al., 1992; Dhomse et al., 2015).

Model simulations are the benchmark method to understand past climate change and attribute the variations seen within observed surface temperature trends (e.g. Hegerl and Zwiers, 2011) to natural and anthropogenic external forcings. Whereas all climate models participating in the 5th and 6th Coupled Model Intercomparison Project (CMIP5 and CMIP6) include interactive aerosol modules for tropospheric aerosol radiative effects, very few use these schemes to simulate the effects of volcanic eruptions. Instead, the climate models performing historical integrations use prescribed volcanic aerosol datasets to mimic climatic effects of the past eruptions. In CMIP5, most climate models used the NASA Goddard Institute for Space Studies (GISS) volcanic forcing dataset (Sato et al., 1993, hereafter, the Sato dataset) that is constructed from SAGE-I, SAM-II and SAGE-II aerosol extinction measurements, combined with an extensive synthesis of pre-satellite-era observational datasets (see https://data.giss.nasa.gov/modelforce/strataer). The Sato dataset consists of zonal-mean stratospheric AOD at 550 nm ($sAOD_{550}$) and column effective radius ($R_{eff}$). The CMIP5 modelling groups used different approaches to apply the 550 nm information across the spectral wavebands of their models' radiative transfer modules and to redistribute the total stratospheric aerosol optical thickness into their model vertical levels (e.g. Driscoll et al., 2012).

Stenchikov et al. (1998) constructed a forcing dataset for Pinatubo that included the variation in aerosol optical properties across wavebands in the SW and LW. They combined SAGE-II and SAM-II (McCormick, 1987) aerosol extinction, as well as infra-red aerosol extinction data from the Improved Stratospheric and Mesospheric Sounder (ISAMS) (Lambert et al., 1993; Grainger et al., 1993; Lambert et al., 1997) and the Cryogenic Limb Array Etalon Spectrometer (CLAES) (Roche et al., 1993). They also compared and/or calibrated to AVHRR, lidar and balloon-borne particle counter observations.

Over the past two decades a large number of chemistry-climate models (CCMs) have been developed, and applied to improve our understanding of past stratospheric change. Several co-ordinated CCM hindcast integrations have been performed via activities such as CCMVal (Eyring et al., 2005, 2008; Morgenstern et al., 2010) and CCMI (Eyring et al., 2013; Morgenstern et al., 2017), with each of the models using different methods to include stratospheric heating from volcanic aerosol clouds. Some CCMs prescribed pre-calculated zonal mean heating rate anomalies (e.g. Schmidt et al., 2006), whilst others applied radiative heating from prescribed aerosol datasets, either the 2-D GISS $sAOD_{550}$ data set or from a 3-D prescribed aerosol surface area density (SAD). SPARC (2010, Chap. 8) analysed lower stratospheric

temperatures following the Pinatubo eruption across different models participating in the CCMVal-2 activity. They showed that CCMVal-2 models show a broad range in the simulated lower stratospheric temperature anomalies (0.5 to 3 K at 50 hPa) with SAD-derived warming tending to be higher than the ~1 K anomaly suggested by ERA-interim reanalysis data.

The other volcanic forcing dataset used in CMIP5 is that from Ammann et al. (2003, hereafter, Ammann dataset), which is based on a parameterisation for the meridional dispersion of volcanic aerosol clouds to mid-latitudes, as determined by the seasonal cycle in the Brewer-Dobson circulation and an assumed 12-month decay timescale for the tropical reservoir. The dataset specifies the forcing for all major tropical eruptions in the 20th century, but does not resolve the effect of the QBO phase in modulating the inter-hemispheric dispersion pathways. The peak aerosol optical depth for each eruption is scaled to match estimates of maximum aerosol loading (Stothers, 1996; Hofmann and Rosen, 1983b; Stenchikov et al., 1998), assuming a fixed particle size distribution ($R_{eff}$ = 0.42 $\mu$m).

For the latest historical simulations in CMIP6 (e.g. Eyring et al., 2016), a single volcanic aerosol dataset was provided for the full 1850-2016 period. This dataset is split into two parts, depending on the availability of satellite data. For 1850-1979 it is based on a simulation of a 2-D interactive stratospheric aerosol model (AER2D) [hereafter CMIP6-AER2D] Arfeuille et al. (2014). For the satellite era (after 1979), the dataset is provided as the Global Space-based Stratospheric Aerosol Climatology (GloSSAC) dataset (Thomason et al., 2018). The combined forcing dataset is designed to enable chemistry-climate models to include aerosol-radiation interactions (aerosol optical properties) consistently along with a prescribed surface area density for heterogeneous chemistry. The aerosol optical properties datasets are tailored for each climate model; the provided properties are mapped onto the model's SW and LW wavebands used in the radiative transfer module (see Luo, 2016).

Here we analyse major volcanic experiments with the interactive stratospheric aerosol configurations of the Unified Model - United Kingdom Chemistry and Aerosol (UM-UKCA) composition-climate model. The model experiments simulate the volcanic aerosol clouds, and associated radiative forcings, from the three largest tropical eruptions over the past century: Mt. Agung (March 1963), El Chichón (April 1982) and Mt. Pinatubo (June 1991). Aligning with the design of the Interactive Stratospheric Aerosol Model Inter-comparison Project (ISA-MIP) co-ordinated multi-model ''Historical Eruption SO$_2$ Emissions Assessment'' (Timmreck et al., 2018), the experiments consist of 3-member ensembles of simulations with each of upper, lower and mid-point estimates of the SO$_2$ emitted from each eruption. We then compare the simulated aerosol properties of the volcanic aerosol clouds to a range of observational datasets.

In addition to the aerosol cloud, the UM-UKCA volcanic experiments include the effects on the stratospheric ozone layer. The GLOMAP-mode aerosol microphysics scheme (Mann et al., 2010; Dhomse et al., 2014) also simulates the tropospheric aerosol layer (Yoshioka et al., 2019), with the stratosphere-troposphere chemistry scheme (Archibald et al., 2020) predicting tropospheric ozone and oxidising capacity consistent with the corresponding decade's composition-climate setting. There have been several improvements to the aerosol microphysics module since our original Pinatubo analysis presented in Dhomse et al. (2014), and these are discussed in Brooke et al. (2017); Marshall et al. (2018, 2019). Section 3 provides the specifics of the model experiments, with Section 4 describing the observational datasets. Model results are given in Section 5. Key findings and conclusions are presented in Section 6.

## 2 Model Experiments

We use the Release Job 4.0 (RJ4.0) version of the UM-UKCA composition-climate model (Abraham et al., 2012), which couples the Global Atmosphere 4.0 configuration (Walters et al., 2014, GA4) of the UK Met Office Unified Model (UM v8.4) general circulation model with the UK Chemistry and Aerosol chemistry-aerosol sub-model (UKCA). The GA4 atmosphere model has a horizontal resolution of $1.875° \times 1.25°$(N96) with 85 vertical levels from the surface to about 85 km. The RJ4.0 configuration of UM-UKCA adapts GA4 with aerosol radiative effects from the interactive GLOMAP

aerosol microphysics scheme and ozone radiative effects from the whole-atmosphere chemistry that is a combination of the detailed stratospheric chemistry and simplified tropospheric chemistry schemes (Morgenstern et al., 2009; O'Connor et al., 2014; Archibald et al., 2020).

The experiment design is similar to that in Dhomse et al. (2014), but with the volcanic aerosol radiatively coupled to the dynamics (as in Mann et al. (2015)) for transient atmosphere-only free-running simulations. Briefly, the model uses

the GLOMAP aerosol microphysics module, and the chemistry scheme applied across the troposphere and stratosphere. Greenhouse gas (GHG) and ozone-depleting substance (ODS) concentrations are from Ref-C1 simulation recommendations in the Chemistry-Climate Model Initiative (CCMI-1; Eyring et al. (2013); Morgenstern et al. (2017)) activity. Simulations are performed in atmosphere-only mode, and we use CMIP6 recommended sea-surface temperatures and sea-ice concentration that are obtained from https://esgf-node.llnl.gov/projects/cmip6/. The main updates since Dhomse et al.

(2014) are: i) updated dynamical model (from HadGEM3-A r2.0 to HadGEM3 Global Atmosphere 4.0), hence improved vertical and horizontal resolution (N48L60 vs N96L85, (Walters et al., 2014)), ii) coupling between aerosol and radiation scheme (Mann et al., 2015), iii) additional sulphuric particle formation pathway via heterogeneous nucleation on transported meteoric smoke particle cores (Brooke et al., 2017). The atmosphere-only RJ4.0 UM-UKCA model used here is identical to that applied in Marshall et al. (2018) and Marshall et al. (2019), with the former run in pre-industrial setting for

the VolMIP interactive Tambora experiment (see Zanchettin et al., 2016) and the latter in year-2000 timeslice mode for a perturbed injection-source-parameter ensemble analysis.

Prior to each of the eruption experiments, we first ran 20-year time-slice simulations with GHGs and ODSs for the corresponding decade (1960 for Agung, 1980 for El Chichón and 1990 for Pinatubo), to allow enough time for the stratospheric circulation and ozone layer to adjust to the composition-climate setting for that time period. Tropospheric aerosol

and chemistry (primary and precursor) emissions were also set to interactively simulate the tropospheric aerosol layer and oxidising capacity for the corresponding decade. For each 20-year time-slice run, we analysed the evolution of stratospheric sulphur burden, ozone, age-of-air and selected long-lived tracers, to check that the model had fully adjusted to the GHG and ODS settings for a given time period. We then analysed timeseries of the tropical zonal wind profile to identify three different model years that gave a QBO transition approximately matching that seen in the ERA-interim reanalysis

(Dee et al., 2011). The initialisation fields for those years was then used to re-start the three ensemble member transient runs. The QBO evolution for each Pinatubo simulation is shown in the Supplementary Material (Figure S1).

For each eruption, a total of nine different volcanically-perturbed simulations were performed, three different "approximate QBO progressions" for each $SO_2$ emission amount (see Table 1). The 9 corresponding control simulations had identical pre-eruption initial conditions and emissions, except that the relevant volcanic emission was switched off. Note that simulated aerosol are not included the calculation of heterogeneous chemistry; the control simulations uses climatological background SAD values in the stratosphere (mean 1995–2006) while the other simulations include effects of associated heterogeneous chemistry via time-varying SAD from Arfeuille et al. (2014).

## 3  Evaluation Datasets

To provide additional context for the UM-UKCA simulated volcanic aerosol clouds, we compare the simulations to three different observation-based volcanic forcing datasets, as well as several individual stratospheric aerosol measurement datasets (see Table 2).

The primary evaluation dataset for this study are the two parts of the volcanic aerosol dataset provided for the co-ordinated CMIP6 historical integrations (Eyring et al., 2016 ). For 1979 onwards, we use CMIP6 recommended GloSSAC (Thomason et al. (2018), hereafter referred to as CMIP6-GloSSAC) dataset. CMIP6-GloSSAC is a best-estimate aerosol extinction dataset from various satellite instruments: SAGE-I, Stratospheric Aerosol Measurement (SAM), the latest version of the SAGE-II dataset and, for the Pinatubo period, the infra-red aerosol extinction measurements from the Halogen Occultation Experiment (HALOE) and Cryogenic Limb Array Etalon Spectrometer (CLAES). Lidar measurements from Hawaii, Cuba and Hampton, Virginia are used to fill the gap in the post-Pinatubo part of the dataset where none of these datasets was able to measure the full extent of the volcanic cloud. For the El Chichón period, airborne lidar surveys between SAGE-1 and SAGE-II period are used. Here we use latest version (V2) of the CMIP6-GloSSAC; key differences between CMIP6-GloSSAC V1 and V2 data are described in Kovilakam et al. (2020).

With the El Chichón eruption occurring between the SAGE-I and SAGE-II instruments, CMIP6-GloSSAC is largely based on combining the SAM-II extinction measurements at 1000 nm (1978–1993), with a 550 nm extinction derived from applying fit to the variation in 550:1020 colour ratio from the SAGE-II period. The SAM-II instrument only measures at high-latitudes, with the period after the El-Chichón eruption (April 1982–October 1984) data constructed via linear interpolation. CMIP6-GloSSAC also uses lidar measurements from the NASA Langley lidar (Hampton, USA) and the 5 aircraft missions after El Chichón: July 1982 (13°N to 40°N), October and November 1982 (45°S to 44°N), January and February 1983 (28°N to 80°N), May 1983 (59°S to 70°N), and January 1984 (40°N to 68°N).

For the Pinatubo period, CMIP6-GloSSAC is an updated version of the gap-filled dataset described in SPARC (2006, Chapter 4), combining SAGE-II aerosol extinction (in the solar part of the spectrum), with HALOE and CLAES aerosol extinction in the infra-red (see Thomason et al., 2018). For the period where the SAGE-II signal was saturated (e.g. Thomason, 1992), CMIP6-GloSSAC applies an improved gap-fill method in mid-latitudes, but in the tropics is still based on the composite dataset from SPARC (2006, pages 140-147), combining with ground-based lidar measurements from

Mauna Loa, Hawaii (19.5°N, (Barnes and Hofmann, 1997)) and, after January 1992, also with lidar measurements from Camaguey, Cuba (23°N, see Antuña, 1996)).

For the period 1984 to 2005, the SAD provided is derived from the SAGE-II multi-wavelength aerosol extinction (Thomason et al., 2008), known as the $4\lambda$ dataset as it uses all 4 aerosol extinction channels (386 nm, 453 nm, 525 nm and 1020 nm). The updated version provided for CMIP6 (the $3\lambda$ dataset) uses only 3 channels as the 386 nm aerosol extinction are excluded due to a higher uncertainty.

For the pre-satellite part of the CMIP6 historical period (1850–1979), the volcanic aerosol properties dataset is constructed using results from the 2-D interactive stratospheric aerosol model (CMIP6-AER2D) and is obtained via ftp: //iacftp.ethz.ch/pub_read/luo/CMIP6/ (last access: January 25, 2020) (Luo, 2016). Each of the volcanic aerosol clouds within the interactive 2-D simulations was formed from individual volcanic $SO_2$ emissions. Each eruption's mass emission and injection height was based on literature estimates, considering also plume-rise model information, and from comparison and calibration to ice core sulphate deposition and ground-based solar radiation measurements (Arfeuille et al., 2014).

Each of these two parts of the CMIP6 dataset (CMIP6-GloSSAC and CMIP6-AER2D) primarily consists of the three parts explained in the Introduction (waveband-mapped aerosol optical properties in the SW and LW, plus surface area density). A 2-D monthly zonal-mean monochromatic aerosol extinction dataset at 550 nm is provided for the full 1850-2014 period, also with monthly zonal-mean effective radius, particle volume concentrations and single-mode log-normal mean radii values. For the CMIP6-GloSSAC part of the dataset, aerosol extinction is also provided at 1020 nm wavelength.

As an extra constraint for the simulated Agung aerosol cloud, we have recovered an important additional observational dataset, which until now has only been available in tables within the appendix of a PhD thesis (Grams, 1966). The dataset provides an important observational constraint to evaluate the progression in the vertical extent of the simulated Agung aerosol cloud. This dataset is the 694 nm backscatter ratio observations from 66 nights of lidar measurements at Lexington, Massachusetts (42°44' N, 71°15' W, Fiocco and Grams, 1964; Grams and Fiocco, 1967) in the periods January to May 1964 (23 profiles), and October 1964 to July 1965 (43 profiles). To enable comparison to the model-predicted 550 nm extinction, the aerosol backscatter ratio observations at 694 nm are converted to aerosol extinction at 532 nm, as described in the Supplementary Material. Note that the Lexington measurements used here are an initial version of a 532 nm extinction profile dataset being prepared for submission to the Earth System Science Datasets journal (Antuna Marrero et al., in prep.) .

To evaluate simulated stratospheric aerosol optical depth (sAOD), we provide three different observation-based datasets to provide greater context for the comparisons. For the CMIP6 dataset, we derive $sAOD_{550}$ by vertically integrating CMIP6-GloSSAC/CMIP6-AER2D 550 nm extinction for all the levels above the tropopause. For the other two volcanic forcing datasets (Sato et al., 1993; Ammann et al., 2003), the $sAOD_{550}$ is specified in the data files, as the primary aerosol metric provided (see Table 2). To analyse the lower stratospheric warming following each eruption, we show the model temperature differences between control and sensitivity simulations alongside the 5-year temperature anomaly from the ERA-Interim reanalysis data (Dee et al., 2011), and overplot the progression of the reanalysis tropical zonal wind profile

to indicate the QBO transitions through each period. For the Agung comparison, we use ERA-40, an earlier 40-year
ECWMF reanalysis dataset (Uppala et al., 2005)

## 4 Results and Discussion

The temporal radiative forcing signature from a major tropical eruption is primarily determined by the evolution of the
volcanic aerosol cloud in the stratosphere. An initial ''tropically confined phase" sees zonally-dispersing $SO_2$ and ash
plume transforming to layered aerosol cloud. Meridional transport in the subsequent "dispersion phase" then leads to
a hemispheric or global cloud of mainly aqueous sulphuric acid droplets. The efficacy of such volcanic clouds' solar
dimming, and the extent of any offset via long-wave aerosol absorption, is strongly linked to how large the sulphuric
aerosol particles grow (their size distribution) as this large-scale dispersion progresses (e.g Lacis et al., 1992).

In the following subsections we assess, for each eruption, the simulated volcanic aerosol cloud for the upper, lower
and mid-point $SO_2$ emissions and compare to available observational constraints. Our focus here is primarily on aerosol
optical properties, evaluating mid-visible stratospheric AOD and aerosol extinction, in both the mid-visible and near-infra-
red, to understand how the altitude and vertical extent of the cloud varies for each eruption. In each case, we also compare
the lower stratospheric warming with the temperature anomaly from the ERA-Interim/ERA-40 reanalyses.

### 4.1 Mt. Pinatubo aerosol cloud

In the Pinatubo case, satellite measurements are able to provide an additional constraint for the particle size evolu-
tion, with particle effective radius derived from the volume concentration and surface area density SAGE-II extinction at
multiple wavelengths (Thomason et al., 1997a; SPARC, 2006). Hence for Pinatubo, we also compare model-simulated
effective radius to that provided with the CMIP6-GloSSAC dataset, which underpins each climate model's specified multi-
wavelength aerosol optical properties in the Pinatubo forcings in CMIP6 historical integrations. With Pinatubo by far the
dominant external forcing in the 1990s, we also compare simulated SW and LW forcings to the Earth Radiation Budget
Experiment (ERBE) satellite data to gain direct insight into how the different $SO_2$ emission simulations evolve in terms of
top-of-the-atmosphere (TOA) radiative forcings.

Baran and Foot (1994) analysed satellite observations of the Pinatubo aerosol cloud from the High-resolution Infrared
Radiation Sounder (HIRS), converting the measured LW aerosol optical properties into a timeseries of global aerosol
burden. In Dhomse et al. (2014), we used this observed global burden dataset to evaluate the model's simulated aerosol
cloud, translating the peak global burden of 19 to 26 Tg from the HIRS measurements into a 3.7 to 6.7 Tg range for
stratospheric sulphur, assuming the particles were 75% by weight aqueous sulphuric acid solution droplets. We identified
an important inconsistency in the model's predictions, when also considering satellite observations of volcanic $SO_2$. The
satellite measurements of $SO_2$ show that 7 to 11.5 Tg of sulphur was present in the stratosphere, for few days after
the eruption (14 to 23 Tg of $SO_2$, Guo et al. (2004a)), so only around 50% of the emitted sulphur remained present at
peak volcanic aerosol loading. In contrast, the model simulations showed that  90% of the sulphur emitted remained

in the volcanic aerosol cloud at its peak global mass burden. This inconsistency was also found in other interactive Pinatubo stratospheric aerosol model studies (Sheng et al., 2015a; Mills et al., 2016), with a number of models finding best agreement with observations for 10 to 14 Tg emitted $SO_2$ (5 to 7 Tg of sulphur), which is less than the lower bound from the TOMS/TOVS measurements. In Dhomse et al. (2014), we suggested the models may be missing some process or influence which acts to redistribute the sulphur within the volcanic cloud, causing it then to be removed more rapidly.

Figure 1a shows the timeseries of global stratospheric aerosol sulphur burden from the present study's Pinatubo simulations, comparing also to the previous interactive Pinatubo UM-UKCA simulations with 20 and 10 Tg $SO_2$ injection as presented in Dhomse et al. (2014). The 20, 14 and 10 Tg $SO_2$ Pinatubo clouds generate a peak loading of 8.3, 5.9 and 4.2 Tg of sulphur, translating into conversion efficiencies of 83, 84 and 84%, respectively. This continuing discrepancy with the satellite-derived 50% conversion efficiency might be due to accommodation onto co-emitted ash particles. Recently we have re-configured the UM-UKCA model to enable new simulations to test this hypothesis (Mann et al., 2019b). We consider the requirement to reduce model-emitted $SO_2$ to be less than that indicated by satellite measurements as an adjustment to compensate for a missing removal/redistribution process in the initial weeks after the eruption.

The simulated Pinatubo global stratospheric sulphur burden in runs **Pin10** and **Pin14** is in good agreement with the HIRS observations, both in terms of predicted peak burden, and the evolution of its removal from the stratosphere. In particular, the model captures a key variation in the HIRS measurements, namely that the removal of stratospheric sulphur was quite slow in the first year after the eruption. The volcanic aerosol cloud retained a steady 4-5 Tg of sulphur for more than 12 months after the eruption, before its removal proceeded at much faster rate in late 1992 and early 1993. The corresponding simulations from Dhomse et al. (2014) (**Pin10** and **Pin20**) show a simpler peak and decay curve, with the removal from the stratosphere proceeding much faster and earlier than the HIRS measurements indicate.

As shown in Mann et al. (2015), and other studies (Young et al., 1994; Sukhodolov et al., 2018), when interactive stratospheric aerosol simulations of the Pinatubo cloud include the heating effect from aerosol absorption of outgoing LW radiation (i.e. the radiative coupling of the aerosol to the dynamics), the resulting enhanced tropical upwelling greatly changes the subsequent global dispersion. In Mann et al. (2015), we also showed that this coupling improves the simulated tropical mid-visible and near infra-red extinction compared to the SAGE II measurements. We identified that the SAGE II measurements are consistent with the combined effects of increased upwelling and later sedimentation, highlighting the need to resolve composition-dynamics interactions. Here we show that this effect also leads to a quite different global sulphur burden, with the later dispersion peak in the mid-latitude sulphur becoming a greater contributor. This behaviour is explored further in Figure 1b, where we assess the e-folding timescale for the removal of stratospheric sulphur, derived by applying a least squares regression fit on 7-month running-mean mass burden values (3 monthly means either side). We find that a Pinatubo realisation that injects more sulphur produces a volcanic aerosol cloud that is removed more rapidly, the effect apparent throughout the decay period. The timing of the accelerating removal occurs consistently across the 3 runs with residence times for **Pin10**, **Pin14** and **Pin20** decreasing from 9, 6 and 4 months in May 1992, to minima of 5, 3 and 2 months in February 1993.

Later (in Figure 4) we assess the behaviour of model-predicted effective radius, showing that it continues to increase steadily in the tropics throughout 1992, the maximum particle size at 20 km occurring in January 1993. That the maximum effective radius occurs at exactly the same time as the minimum in e-folding time illustrates the importance for interactive stratospheric aerosol models to represent its increased size, sedimentation of particles proceeding faster as the particles grow larger. One thing to note however, is that although the different volcanic $SO_2$ amount is emitted at the same altitude,
since the runs are free-running, later we show that each different emission amount causes different amounts of heating, the resulting enhancements to tropical upwelling lofting the cloud to different altitudes.

   The predicted stratospheric sulphur burdens in **Pin10** and **Pin14** compares well to the observations, suggesting a 10 Tg to 14 Tg $SO_2$ emission range will produce a volcanic aerosol cloud with realistic volcanic forcing magnitude. The comparison could provide a test for other interactive stratospheric models, to identify a model-specific source parameter
calibration. It should be noted that such a reduction in emissions, to values below the $SO_2$ detected (Guo et al., 2004a), is a model adjustment, likely compensating for a missing sulphur loss/re-distribution process.

   We also note some differences in sulphur burden between this study's interactive Pinatubo simulations and the previous equivalent simulations presented in Dhomse et al. (2014). Firstly, the background burden in run **Pin00** is much lower (0.11 Tg) than previous simulations (0.50 Tg) and now in reasonable agreement with other studies (Hommel et al., 2011; Sheng
et al., 2015b; Kremser et al., 2016), and at the lower end of the burden range estimates in SPARC (2006) of 0.12-0.18 for Laramie optical particle counter (OPC) balloon soundings and 0.12-0.22 Tg Garmisch lidar measurements, respectively. They are reported as 0.5-0.7 Tg and 0.5-0.9 Tg mass of 75% weight aqueous sulphuric acid solution, respectively. The main reason for the reduction in simulated quiescent stratospheric sulphur burden, compared to Dhomse et al. (2014), is the influence from meteoric smoke particles (MSP), forming meteoric-sulphuric particles (Murphy et al., 2014). One of
the effects from simulating these particles, alongside homogeneously nucleated pure sulphuric acid particles, is also to reduce the sulphur residence time, compared to equivalent quiescent simulations with pure sulphuric particles only (Mann et al., 2019a). There are also some dynamical differences in the updated simulations here, which use an improved vertical and horizontal resolution model (N96L85 rather than N48L60), that might influence stratosphere-troposphere exchange and stratospheric circulation (e.g. Walters et al., 2014).

Secondly, we also assess the simulated stratosphere into the 3rd post-eruption year (after June 1993). Although for the first two years, the model's global stratospheric sulphur in the simulations **Pin10** and **Pin14** tracks closely with HIRS estimates (Figure 1a), the satellite-derived S-burden drops off rapidly from about 3 Tg in January 1993 to 0.5 Tg by September 1993. On the other hand, the simulated volcanic aerosol cloud does not disperse down to that value until September 1994. However, this accelerated loss of stratospheric sulphur in the HIRS data seems to be partially consistent
with other satellite measurements, for example SAGE-II measurements (see Figure 3), as well as OPC measurements (Thomason et al., 1997b) and CLAES observations (e.g. Bauman et al., 2003; Luo, 2016). This suggests that latter part of the HIRS data may well be accurate, though it seems difficult to identify a driving mechanism for this. Each of the model experiments suggest the stratospheric aerosol remained moderately enhanced throughout 1993 and 1994.

Figure 2 shows, for each eruption magnitude, the zonal mean ensemble-mean stratospheric AOD at 550 nm ($sAOD_{550}$) from the UM-UKCA Pinatubo simulations (**Pin10**, **Pin14**, **Pin20**), that are compared to three different volcanic forcing datasets. To clarify the exact nature of the easterly QBO phase and sAOD evolution in each ensemble member, these are shown in the Supplementary Figures S2 and S3, respectively. For this period, the CMIP6-GloSSAC V2 data should be considered the primary one, as it is based on the latest versions of each of the different satellite products (Thomason et al., 2018; Kovilakam et al., 2020).

As in the HIRS sulphur burden comparisons (Figure 1), the **Pin20** simulation, which best matches the satellite-observed $SO_2$ estimates, strongly over-predicts the stratospheric AOD in the tropics and Northern Hemisphere (NH) mid-latitudes, compared to all three reference datasets. However, whereas the lower emissions runs **Pin10** and **Pin14** both closely track the observed global column sulphur variation, run **Pin10** has best agreement with all three reference datasets for mid-visible sAOD. For this run **Pin14** is high-biased in the tropics and NH mid-latitudes. In the tropics, all three emission-magnitude ensembles are higher than the reference datasets.

Figure 2 illustrates the well-established global dispersion pattern for the Pinatubo aerosol cloud: initially confined to the tropical reservoir region, then dispersing to mid-latitudes, following the seasonal variation in the Brewer-Dobson circulation. The over-prediction in the tropics is a common feature among interactive stratospheric aerosol models. It is noticeable that this over-prediction is worst in the first 6-9 months after the eruption, which could indicate the source of the model's discrepancy. Whereas an overly non-dispersive tropical pipe in the model could be the cause, the timing is potentially more consistent with a missing loss pathway that is most effective in the initial months after the eruption. Co-emitted volcanic ash will also have been present within the tropical reservoir, as seen in the airborne lidar depolarisation measurements in the weeks after the eruption (Winker and Osborn, 1992), and remained present in the lowermost part of the mid-latitude aerosol cloud in both hemispheres (Young et al., 1992; Vaughan et al., 1994). The AOD high bias is consistent with the hypothesis that a substantial proportion of the emitted sulphur may have been removed from the stratosphere by accommodation onto the sedimenting ash. If this mechanism is causing such a vertical re-distribution within the tropical reservoir, it will increase the proportion of Pinatubo sulphur being removed into the troposphere via the rapid isentropic transport that occurred during the initial months in the lowermost stratosphere. Furthermore, stratospheric AOD is not a measure of sulphur, and the variations in sAOD will partly indicate changes in scattering efficiency that results from the gradient in effective radius that were apparent at the time, as discussed in this section.

The peak mid-visible AOD from AVHRR is higher than the SAGE-II gap-filled satellite measurements (Long and Stowe, 1994). For example, as noted in Thomason et al. (2018), the peak mid-visible stratospheric AOD in the AVHHR dataset is around 0.4, compared to 0.22 in GloSSAC. However, other possible model biases cannot be ruled out. One consideration for these free-running simulations, even with each ensemble member initialised to approximate the period's QBO phase, is that nudging towards reanalysis meteorology would give more realistic representation of this initial phase of the plume dispersion (Sukhodolov et al., 2018). We chose to perform free-running simulations to allow the enhanced tropical up-welling resulting from increased LW aerosol-absorptive heating, consistent with the $SO_2$ emission, known to exert strong influence on the subsequent simulated global dispersion (Young et al., 1994).

In contrast to the tropics and NH mid-latitudes, where run **Pin10** agrees best with the reference datasets, run **Pin14** compares best to the Southern Hemisphere (SH) $sAOD_{550}$ measurements in GloSSAC. This difference may be highlighting the requirement for a more accurate simulation of the QBO evolution, likely necessary to capture the Pinatubo cloud's transport to the SH mid-latitudes (e.g. Jones et al., 2016; Pitari et al., 2016). One thing to note is that our simulations do not include the source of volcanic aerosol from the August 1991 Cerro Hudson eruption in Chile. However, measurements from SAGE-II (Pitts and Thomason, 1993) and ground-based lidar (Barton et al., 1992) indicate that the Hudson aerosol cloud only reached to around 12 km, with the Pinatubo cloud by far the dominant contributor to SH mid-latitude sAOD. So, although we have not included the Hudson aerosol in our simulations, hence it is possible that minor contributor from Cerro Husdson sAOD might reduce differences between **Pin10** and CMIP6-GloSSAC V2 sAOD. Overall, the $sAOD_{550}$ comparisons confirm the findings from Figure 1 that for UM-UKCA, consistent with other global microphysics models (Sheng et al., 2015a; Mills et al., 2016), Pinatubo aerosol properties are better simulated (acknowledging the discrepancy in the SH) with a 10 Tg to 14 Tg range in volcanic $SO_2$ emission.

Although Figure 2 suggests significant differences among the volcanic forcing datasets for the Pinatubo period, the CMIP6-GloSSAC data is the reference dataset while the 1991-94 period in the Sato dataset is mostly based on an earlier version of the SAGE-II data. The GloSSAC data have been compared extensively with lidar measurements (Antuña et al., 2002; Antuña, 2003), and combined for the gap-filled dataset (SPARC, 2006) with improvements in the SAGE-II aerosol extinction retrieval algorithm (version 7).

For historical climate integrations in CMIP5, some models used the Sato forcing dataset whilst others used Ammann and their differences affect interpretation of volcanic impacts among the models (Driscoll et al., 2012). For CMIP6, all models have harmonised to use the same forcing dataset, with a dedicated VolMIP analysis to compare the climate response in each model and with the CMIP6-GloSSAC Pinatubo forcing applied to the pre-industrial control (Zanchettin et al., 2016).

After comparing the total sulphur burden and sAOD, Figure 3 shows UM-UKCA simulated mid-visible extinction at 3 different altitudes in the lower stratosphere, to evaluate the simulated vertical extent of the Pinatubo cloud through the global dispersion phase. For the tropics, extinction comparisons are shown at 24 km, 28 km and 32 km, whereas for SH (35°S-60°S) and NH (35°N-60°N) mid-latitudes the chosen levels are 20 km, 24 km and 28 km, to account for the higher tropical tropopause. Simulated extinctions are compared with raw SAGE v7.0 data (Damadeo et al., 2013) as well as the gap-filled extinction from CMIP6-GloSSAC at 525 nm. As discussed previously, extinctions from **Pin14** (and to some extent **Pin10**) show much better agreement with observational data for all three latitude bands. Most importantly, model extinction remain close or slightly lower in the mid-latitude compared to SAGE II extinction even after 4 years, suggesting that the sharp decay in sulphur burden observed by Baran and Foot (1994) may be unrealistic. Interestingly, in the SH mid latitudes, extinction from **Pin14** shows much better agreement with SAGE II extinctions at 20 and 24 km. This again confirms biases discussed in Figure 2 that could be attributed to the weaker lower stratospheric transport in the SH mid-latitudes, and Cerro Hudson eruption must have only slight contribution to the sAOD. At 1020 nm, agreement is even better (See Supplementary Figure S3). Also as observed in Figure 1 and 2, extinction differences between runs

**Pin10**, **Pin14** and **Pin20** are largest for the first few months after the eruption but extinction lines almost overlap within ensemble variance from each eruption. This again confirms that as greater amount of volcanic $SO_2$ is injected into the stratosphere, the cloud evolves to a larger average particle sizes, leading to faster sedimentation.

A key feature seen in Figure 3 that is not captured well in any Pinatubo simulation is the plateau in the SAGE-II (and GloSSAC) tropical peak extinction. For example, at 24 km (where the instrument saturation effect should be minimal), after reaching peak values within first three months, extinction values remain approximately constant for at least 6 months. At 20 km, this plateau in extinction in the tropics is visible for almost 12 months in the CMIP6-GloSSAC data (not shown). Similar features are visible at 1020 nm extinction (Figure S3). If indeed these plateau features in the SAGE-II data are realistic, then they would need to have been caused by the sustained tropical upwelling (via upward branch of Brewer-Dobson circulation, combined with aerosol-induced heating), being offset by sedimentation of the particles that would have grown via condensation and coagulation. These plateau structures in extinctions are not apparent in the mid-latitudes of either hemisphere, with a clear seasonal cycle occurring due to preferential wintertime circulation (e.g. Dhomse et al., 2006, 2008), that is visible in both model and SAGE-II data.

A notable discrepancy is that modelled extinction is low-biased (by up to 50%) during pre-eruption months. This could be associated with low background sulphur burden in our model or slightly elevated stratospheric aerosol due to small volcanic eruptions (such as Mt. Redoubt 1989/90, Kelud,1990) that are not included in our simulations. Another explanation could be due to the fact that the model does not resolve the uptake of organics into the particle phase. Observations (Murphy et al., 2007) and modelling studies (Yu et al., 2016) have shown that organic-sulphate particles (Murphy et al., 2014) are the dominant aerosol type in the tropical and mid-latitude upper troposphere and lower stratosphere, and the omission of this interaction might have introduced systematic low bias during background periods.

Next we evaluate the meridional, vertical and temporal variations in effective radius ($R_{eff}$) in the Pinatubo UM-UKCA datasets. The particle size variations in these interactive simulations of the Pinatubo cloud reflect the chemical and microphysical processes resolved by the chemistry-aerosol module, in association with the stratospheric circulation and dynamics occurring in the general circulation model. We analyse these model-predicted size variations alongside those in the benchmark observation-based $R_{eff}$ dataset from CMIP6-GloSSAC, which applies the 3-$\lambda$ size retrieval from the 453 nm, 525 nm and 1020 nm aerosol extinction measurements from SAGE-II (Thomason et al., 1997a, 2018).

Figure 4 shows zonal mean $R_{eff}$ at 25 km, within the altitude range of the volcanic $SO_2$ injection, and at 20 km, underneath the main volcanic cloud. Results are shown from 3-member means from the 10, 14 and 20 Tg $SO_2$ emission runs (**Pin10**, **Pin14** and **Pin20**). For comparability with the equivalent figure from Dhomse et al. (2014), the Supplementary Material (Figure S6) shows the updated comparison to the Bauman et al. (2003) $R_{eff}$ dataset. Overall, the model captures the general spatio-temporal progression in the $R_{eff}$ variations seen in the GloSSAC dataset. However, whereas the 10 Tg and 14 Tg simulations agree best with the HIRS-2 sulphur burden (Figure 1) and the GloSSAC sAOD and extinction (Figures 2 and 3), the magnitude of the $R_{eff}$ enhancement is best captured in the 20 Tg run (**Pin20**). The comparisons suggest the low bias in simulated $R_{eff}$ seen in the previous UM-UKCA Pinatubo study (Dhomse et al., 2014) is still present here. However, this low bias in particle size/growth may simply be reflecting the required downward-adjustment

of the Pinatubo $SO_2$ emission, as a larger $R_{eff}$ enhancement in the 20 Tg simulation is clearly apparent. It is possible
that the two-moment modal aerosol dynamics in GLOMAP-mode may affect its predicted $R_{eff}$ enhancement. However,
the model requirement for reduced $SO_2$ emission is attributed to be likely due to a missing, or poorly resolved, model
loss pathway, such as accommodation onto co-emitted volcanic ash. The sustained presence of ash within the Pinatubo
cloud (e.g. Winker and Osborn, 1992) will likely have altered particle size and growth rates in the initial months after the
eruption.

In the tropics, where $R_{eff}$ increases are largest, the timeseries of $R_{eff}$ is noticeably different in the core of the tropical
reservoir (10°S to 10°N) to that in the edge regions (10°N-20°N and 10°S-20°S), at both 20 km and 25 km. The $R_{eff}$
increases in these edge regions occur when tropics to mid-latitude transport is strongest, in phase with the seasonal
cycle of the Brewer-Dobson circulation, which tends to transport air towards the winter pole (Butchart, 2014). The $R_{eff}$
increases are due primarily to particle growth from coagulation and condensation, and the simulations also illustrate how
the simulated Pinatubo cloud comprises much smaller particles at 25 km than at 20 km. The 25 km level is in the central
part of the Pinatubo cloud, particles there being younger (and smaller) because the oxidation of the emitted volcanic $SO_2$
that occurs at that level triggers extensive new particle formation in the initial months after the eruption (e.g. Dhomse
et al., 2014). By contrast, at the 20 km level particles will almost exclusively have sedimented from the main cloud, and
therefore be larger. There is a slow but sustained increase in average particle size in the equatorial core of the tropical
Pinatubo cloud, with the 20 km level reaching peak $R_{eff}$ values only during mid-1992, in contrast to the peak sulphur
burden and $sAOD_{550}$ which have already peaked at this time, being in decay phase since the start of 1992 (see Figures
1 and 2).

Whereas the simulated peak $R_{eff}$ enhancement occurs by mid-1992 in the tropics, the peak $R_{eff}$ in NH mid-latitudes
occurs at the time of peak meridional transport, the $R_{eff}$ variation there reflecting the seasonal cycle of the Brewer-Dobson
circulation, as also seen in the tropical reservoir edge region. The different timing of the volcanic $R_{eff}$ enhancement in
the tropics and mid-latitudes is important when interpreting or interpolating the in-situ measurement record from the
post-Pinatubo OPC soundings from Laramie (Deshler, 2003). Russell et al. (1996) show that the $R_{eff}$ values derived
from Mauna Loa ground-based remote sensing are substantially larger than those from the dust-sonde measurements
at Laramie. The interactive Pinatubo simulations here confirm this expected meridional gradient in effective radius, with
the chemical, dynamical and microphysical processes also causing a vertical gradient in the tropical to mid-latitude $R_{eff}$
ratio. The current ISA-MIP activity (Timmreck et al., 2018) brings a potential opportunity to identify a consensus among
interactive stratospheric aerosol models for the expected broad-scale spatio-temporal variations in uncertain volcanic
aerosol metrics such as effective radius.

An important aspect of volcanically enhanced stratospheric aerosol is that they provide surface area for catalytic ozone
loss (e.g. Cadle et al., 1975; Hofmann and Solomon, 1989). A cmparison of stratospheric sulphate area density for three
different months (December 1991, June 1992 and December 1992) is shown in Figure 5. SAD derived using observational
data (Arfeuille et al., 2014), also known as $3\lambda$ SAD, is also shown. Again, **Pin20** SAD shows a high bias, whereas **Pin10**
SAD seems to show good agreement with $3\lambda$ data. Our simulations do not include the $SO_2$ injection from the August

1991 Cerro Hudson eruption (Chile), yet the model captures the volcanic SAD enhancement in the SH mid-latitude stratosphere very well. The model does not capture the enhanced SAD signal at 10-12 km in the SH in December 1991, the altitude of that feature in the 3$\lambda$ dataset is consistent with lidar measurements of the Hudson aerosol cloud from Aspendale, Australia (Barton et al., 1992). The most critical differences are that 3$\lambda$ SAD are confined in the lowermost stratosphere. A deeper cloud of enhanced SAD, with steeper low-high latitude SAD gradients, is visible in all the model simulations. As seen in Figure 3, by June 1992 tropical SAD from runs **Pin10** and **Pin14** are low-biased, indicating lower aerosol in the tropical pipe which could be either due to faster transport to the high latitudes (weaker subtropical barrier in the middle stratosphere) and/or quicker coagulation causing faster sedimentation.

Figure 6 shows the time series of observed SW and LW radiative near-global mean flux anomalies (60°S - 60°N), with respect to a 1985 to 1989 (pre-Pinatubo) baseline. The ERBE data (black symbols) is from Edition 3 Revision 1, non-scanner, wide field-of-view observations (Wielicki et al., 2002). Coloured lines indicate ensemble mean forcing anomalies from three Pinatubo SO$_2$ emission scenarios. The Pin10 simulation generates a peak solar dimming of 4 W/m2, matching well both the timing and magnitude of the peak in the ERBE SW anomaly timeseries. It is notable that if the ERBE SW anomaly is calculated relative to the 1995-1997 baseline, we estimate a peak solar dimming of 5.5 W/m$^2$ (not shown), which then compares best with the Pinatubo SW forcing from Pin14. Consistent with the sulphur burden, sAOD550 and mid-visible extinction comparisons (Figures 1, 2 and 3), the Pin20 simulation also over-predicts the magnitude of the Pinatubo forcing compared to the ERBE anomaly. It is important to note here that the model Pinatubo forcings are not only from the volcanic aerosol cloud, but include also any effects from the simulated post-Pinatubo changes in other climate forcers (e.g. stratospheric ozone and water vapour). As expected run **Pin20** shows largest anomalies in both SW and LW radiation and distinct differences between **Pin10**, **Pin14** and **Pin20** are visible until the end of 1992. For this 10 to 20 Tg emission range, we find the global-mean SW forcing scales approximately linearly with increasing SO$_2$ emission amount, the 40% increase from 10 to 14 Tg and 43% increase from 14 to 20 Tg causing the Pinatubo SW forcing to be stronger by 34% (4.1 to 5.5 W/m$^2$) and 36% (5.5 to 7.5 W/m$^2$), respectively.

In contrast to the SW forcings, the magnitude of the anomaly in the peak LW forcing is best matched in the **Pin20** simulation, although the **Pin14** simulation also agrees quite well with the ERBE anomaly timeseries. Whereas the Pinatubo SW forcing closely follows the mid-visible aerosol changes, the LW forcing is more complex to interpret. Simulated LW aerosol absorption is not studied in this paper and almost certainly has a different temporal variation than the 550 nm and 1020 nm extinction variations analysed here. Also, the model LW forcing also includes effects from the dynamical changes in stratospheric water vapour which partially offset the SW dimming (e.g. Joshi et al., 2003) adding to the LW aerosol effect. Our simulations do not include co-emission of water vapour, which might have influenced stratospheric chemistry (e.g. LeGrande et al., 2016) and altered observed Pinatubo forcing. Another possible explanation for this discrepancy might be a poorly sampled signal in LW radiation alongside ERBE temporal coverage (36 days vs 72 days). Again, as in the sulphur burden and extinction comparisons, after January 1992 observed SW anomalies seem to decay at a faster rate compared to all the model simulations.

Another important volcanic impact is the aerosol-induced heating in the lower stratosphere as large particles absorb outgoing LW radiation. Since the ERA-interim analysis assimilates radiosonde observations from large number of sites in the tropics, we can compare the temperature anomaly to the model predictions, as a further independent test. However, exact quantification of this mechanism is somewhat complicated as the ERA-Interim stratospheric temperature anomalies also include influence from other chemical and dynamical changes such as variations in ozone and water vapour as well as QBO and ENSO-related changes in tropical upwelling (e.g. Angell, 1997b; Randel et al., 2009). We can assume the 5-year anomalies will remove effects of some of the short-term processes. Figure 7 shows ERA-interim temperature anomalies compared to the model's volcanic warming, i.e. the temperature difference between the sensitivity (**Pin10**, **Pin14** and **Pin20**) and control (**Pin00**) simulations. Although we compare the simulated Pinatubo warming (temperature difference) to ERA-interim temperature anomalies, it is only intended to provide an approximate observational constraint for the magnitude of the effect and the altitude at which it reaches a maximum. The **Pin10** simulation best captures the magnitude of the ERA-Interim post-Pinatubo tropical temperature anomalies, and the model simulations and reanalysis both show maximum warming occurred in the 30 to 50 hPa range around 3-4 months after the eruption. The model predicts that the Pinatubo aerosol cloud continued to cause a substantial warming ($> 2$ K) throughout 1992, which propagates downwards as found in ERA-Interim temperature anomalies.

### 4.2 El Chichón aerosol cloud

Whereas Pinatubo is often the main case study to evaluate interactive stratospheric aerosol models, El Chichón provides a different test as its volcanic aerosol cloud dispersed almost exclusively to the NH. We therefore seek to understand whether the biases seen for Pinatubo (over-predicted tropical sAOD and discrepancy between literature estimates of $SO_2$ emission and the peak global aerosol loading) are also seen for this alternative major eruption case.

Both El Chichón and Pinatubo eruptions occurred in the modern satellite era, however there are far fewer datasets available for the evaluation of El Chichón aerosol properties as it occurred in the important gap period between SAGE-I and SAGE II (see Thomason et al., 2018). As there are quite extensive observational data records for the El Chichón volcanic aerosol clouds (e.g McCormick and Swissler, 1983; Hofmann and Rosen, 1983a), combining these with satellite datasets would greatly reduce large uncertainties concerning the evolution of the El-Chichón aerosol cloud (e.g. Sato et al., 1993; SPARC, 2006).

Here, our analysis focuses primarily on comparing simulated mid-visible stratospheric AOD at 550 nm ($sAOD_{550}$) to the CMIP6 and Sato datasets. We also test the simulated vertical extent of the El Chichón cloud, comparing extinction at 20 km and 25 km to the SAGE II (and GloSSAC) data record, and compare the model simulated warming in the tropical lower stratosphere to temperature anomalies in the ERA-Interim reanalyses.

Figure 8 compares ensemble mean $sAOD_{550}$ from **Elc05**, **Elc07**, **Elc10** and three observation-based datasets. Overall, there are significant differences between simulated $sAOD_{550}$ and the observations. The CMIP6-GloSSAC dataset enacts strongest solar dimming in NH mid-latitudes (peak $sAOD_{550}$ of 0.14), the tropical reservoir never exceeding a sAOD of 0.08, whereas the Sato and Ammann datasets, enact highest $sAOD_{550}$ in the tropics. The model simulations also find

highest solar dimming occurred in the tropical reservoir, with the mean of the 5 Tg simulations predicting a maximum sAOD$_{550}$ of about 0.28. With the QBO in the westerly phase, and timing of the eruption (4th April), the Brewer-Dobson circulation readily exported a large fraction of the plume to the NH, but the meridional gradient in the solar dimming is an important uncertainty to address in future research

In the model the depth of the tropical volcanic aerosol reservoir that forms is closely linked to the altitude of the volcanic SO$_2$ emission. We aligned our experiments with the ISA-MIP HErSEA experiment design (Timmreck et al., 2018), specifying a 24-27 km injection height based on the airborne lidar measurement surveys of the tropical stratosphere that provide the main constraint for the gap-filled dataset (see Figure 4.34 in SPARC (2006)). Balloon measurements from southern Texas and Laramie (Hofmann and Rosen, 1983b), and the constraints from the airborne lidar survey flights in July, September and October (McCormick and Swissler, 1983), together show that a large part of the plume was transported early to NH mid-high latitudes via the middle branch of the Brewer-Dobson circulation to around 25 km, with lower altitudes of the cloud remaining confined to the tropical reservoir. The evolution of the cloud is complex and strongly influenced by several factors, including the rate of SO$_2$ conversion to aerosol and the depletion of oxidants, the tropical upwelling of the Brewer-Dobson circulation, sedimentation of the ash and sulphuric acid droplets (and their interactions) and the downward propagating QBO. The multiple interacting processes within the tropical reservoir makes analysis of this early phase dispersion a complex problem, yet their combined net effects determines the subsequent transport of the aerosol to mid-latitudes, and the resulting radiative forcing.

Due to significant differences observed in Figure 8, even with limited SAGE-II observations, simulated extinctions are compared in Figure 9. Simulated extinctions for all three SO$_2$ emission scenarios show an excellent agreement with SAGE-II from October 1984 onwards. Similarly, extinction at 1020 nm also shows very good agreement with SAGE-II data that is shown in Supplementary Figure S4. A sudden jump in the CMIP6-GloSSAC data at the start of the SAGE-II period is evident, and other unexplained sudden increases in extinction earlier in the CMIP6 dataset, e.g. in the SH at 24 km. On the other hand, the somewhat elevated SAGE-II extinction in the NH mid-latitudes compared to model extinctions highlights a possible model discrepancy due to the injection altitude leading to faster removal of the aerosol particles. GloSSAC extinction in the SH mid-latitude shows very little seasonal variation, and the sudden changes seen at both 20 and 24 km are surprising and difficult to reconcile with expected variation. They could potentially be artefacts from the interpolation procedure. Overall, Figure 9 clearly suggests potential areas where combining with models may help improve the CMIP6-GloSSAC (and other) datasets, highlighting the need for combining observational data with El Chichón-related model simulations to better represent the consistency and variations within the El Chichón surface cooling included in climate models.

Figure 10 shows the tropical warming of the stratosphere predicted by the model, comparing again to the ERA-Interim temperature anomaly (compared to the mean for 1982-1986). As in the Pinatubo case (Figure 7), the speed of downward propagation of these anomalies seems to be well captured by all the simulations. Peak warming of about 3 K observed in ERA-Interim between 30-50 hPa seems to be well reproduced in **Elc07**. Warm anomalies (up to 1 K) visible in ERA-Interim data between 10-20 hPa suggest the downward propagating westerly QBO contributed to up to 1 K warming,

hence the simulated warming will be about 1 K less than the ERA-Interim anomalies. Overall, **Elc05** seems to reproduce the El Chichón-related warming more realistically but the slight warming persisting near 70 hPa until March 1983 is absent in this simulation. Again this suggests that for UM-UKCA, 5 Tg and 7 Tg are reasonable lower and upper limits of $SO_2$ injection required to simulate observed lower stratospheric warming.

### 4.3 Mt. Agung aerosol cloud

The El Chichón and Pinatubo eruptions occurred when satellite instruments were monitoring the stratospheric aerosol layer, and the global dispersion of their volcanic aerosol clouds are relatively well characterised. For the Agung period our knowledge of the global dispersion is less certain and primarily based on the synthesis of surface radiation measurements from Dyer and Hicks (1968). These measurements show the Agung cloud dispersed mainly to the SH, although aerosol measurements from 10 balloon-borne particle counter measurements from Minneapolis in 1963-65 (Rosen, 1964, 1968) and ground-based lidar from Lexington, Massachusetts in 1963 and 1964 (Grams and Fiocco, 1967) show substantial enhancement in the NH as well. For this period, the Sato forcing dataset enacts solar dimming following the ground-based solar radiation measurements discussed in Dyer and Hicks (1968), whereas the CMIP6-AER2D dataset is based on a 2-D interactive stratospheric aerosol model simulations.

To address these limitations, the SPARC (Stratosphere-Troposphere Process and their Role in Climate Project) project entitled SSiRC (Stratospheric Sulphur and its Role in Climate) initiated a stratospheric aerosol data rescue project (see http://www.sparc-ssirc.org/data/datarescueactivity.html). Its primary aim is to gather and in some cases re-calibrate post-Agung aerosol measurements from major volcanic periods to provide new constraints for stratospheric aerosol models. For example, ship-borne lidar measurements of the tropical volcanic aerosol reservoir after Pinatubo have recently been recovered (Antuna-Marrero et al., 2020; Mann et al., 2020). As part of this study, we are contributing to this SSiRC activity and have recovered the Lexington post-Agung ground-based lidar measurement from Grams and Fiocco (1967) and use these to constrain the vertical extent of the Agung aerosol cloud.

Figure 11 compares $sAOD_{550}$ from model simulations with CMIP6-AER2D, Sato and Ammann data. Both CMIP6 and Sato datasets suggest that the tropical volcanic aerosol cloud dispersed rapidly, and almost exclusively, to the SH which is also consistent with our understanding of QBO-dependent meridional transport (Thomas et al., 2009). This means that during the westerly QBO phase the volcanic plume is quickly transported towards the winter hemisphere whereas during the easterly phase the tropics-to-high-latitude transport is slower, hence some part of the plume is available for the wintertime transport into the opposite hemisphere. In contrast, the Ammann dataset suggests a significant part of the cloud was transported to the NH, the dispersion parameterisation considering only seasonal changes in stratospheric circulation. Hence, the modulation of meridional transport caused by the QBO, in the Agung case, increasing the export from low to mid-latitudes, is not represented in the Ammann dataset.

Figure 11 also shows that for the post-Agung period there are very large differences in the $sAOD_{550}$ between CMIP6-AER2D and Sato data. Hence, the historical climate integrations performed using these two forcing datasets will have simulated substantially different Agung surface cooling between the two CMIP assessments. Overall, the CMIP6 dataset

generates much stronger peak $sAOD_{550}$ than the Sato dataset, with a peak of around 0.2 in the tropics, just a few months after the eruption. The Sato dataset shows a peak value of about 0.12, which suddenly drops below 0.05 within couple of months. Thereafter, there is a steady build-up with a local peak in $sAOD_{550}$ occurring in November 1963, 8 months after the eruption. The Sato dataset then enacts a much stronger second peak in tropical $sAOD_{550}$ in August-September 1964

that must be based on measurements from Kenya and the Congo (Dyer and Hicks, 1968). By contrast, CMIP6-AER2D, predicts the Agung cloud dispersed rapidly to the SH with the tropical reservoir reducing to $sAOD_{550}$ of less than 0.05 at that time. Our simulations predict the Agung aerosol dispersed to the SH with similar timing to the CMIP6 dataset, but with a larger proportion remaining in the tropical reservoir. Similar to the CMIP6 datasets, our simulations also predict a secondary $sAOD_{550}$ peak in SH mid-latitudes near $40°$S. Although a similar pattern is produced in almost all simulations,

$sAOD_{550}$ from **Agu06** seems to be in much better agreement with CMIP6 data.

    These comparisons highlight that there is still substantial uncertainty about the global dispersion of the Agung cloud. However, there are extensive set of stratospheric aerosol measurements carried out during this period (see http://www. sparc-ssirc.org/data/datarescueactivity.html). Hence, there is potential to reduce this uncertainty combining these observations also with interactive stratospheric model simulations (Timmreck et al., 2018). Dyer and Hicks (1968) discuss the

transport pathways for the volcanic aerosol, in relation to seasonal export from the tropical reservoir. Stothers (2001) analysed a range of measurements to derive the turbidity of the Agung cloud, but they neglected measurement from Kenya and Congo sites in their analysis, attributing a lower accuracy in those data. It is notable that those observations were during the dry season, when other sources of aerosol could potentially have caused additional solar dimming. In terms of modeling, Niemeier et al. (2019) discussed possible implications of two separate Agung eruptions in 1963. They per-

formed two model simulations, one with a single eruption and one with two separate eruptions on 17th March and 16th May with 4.7 Tg and 2.3 Tg $SO_2$ injection, respectively. They found significant differences between simulated aerosol properties and available evaluation datasets. They suggested that two separate eruptions are necessary to simulate the climatic impact. However, due to limited observational data they could not validate their model results extensively. They also discussed that simulated $sAOD_{550}$ differences with respect to evaluation data are larger than the differences between

their two simulations. Pitari et al. (2016) also present global mean $sAOD_{550}$ changes after the Agung eruption with single eruption (12 Tg on 16 May 1963), but they did not show the latitudinal extent of the Agung volcanic cloud dispersion.

    Figure 12 compares simulated and CMIP6-AER2D extinctions at 550 nm at 16, 20 and 24 km. As in previous figures the tropical comparison is shifted upwards by 4 km. Overall, modelled and CMIP6 extinctions show an almost identical decay rate. At 16 km, nearly all the model simulations show a high bias compared to CMIP6 data and model extinction.

On the other hand, at 20 km, tropical CMIP6 extinction seems to peak a bit later and there is better agreement in the mid-latitude extinction in both hemispheres. The UM-UKCA extinctions reflect the primary influence from the QBO because of changing the sub-tropical edge of the tropical reservoir as well as peak wintertime meridional transport in either hemisphere. On the other hand, CMIP6 extinctions show a strong seasonal cycle in the tropics. The differences between our model and CMIP6 extinction must be primarily due to injection altitude and the simplified aerosol microphysical model

used to construct CMIP6 data.

Figure 12 also shows the extinction from the early ground-based lidar at Lexington, Massachusetts (42°44' N, 71°15' W) as presented in Grams and Fiocco (1967). The method used to convert lidar backscatter to extinction is described in the Supplementary Material. Although the lidar data shows large variability, these single location measurements still provide better insight into the transport of the Agung aerosol cloud in the NH. At 16km, **Agu09** seem to show better agreement with lidar data, although by spring 1965, simulated extinctions are lower than the lidar data, suggesting faster decay in the model at this level. A similar pattern is observed at 20 km. Somewhat larger lidar extinction in spring 1965 compared to model simulations might be due to either weak transport from the tropics to NH mid-latitude (more transport to the SH) in the model, or aerosol removal is too fast in our simulations. Extinctions at 24 km are shown in the Supplementary Figure S5, and again confirm good agreement between lidar and **Agu09**. Overall, the extinction comparison with the Lexington lidar data suggests that transport of the Agung volcanic cloud and its vertical extent in to the NH mid-latitude is well represented in **Agu09**.

Finally, we compare tropical warming in Figure 12. As ERA-Interim reanalyses starts in 1979, we calculate observational-based anomalies from ERA–40 data. Bearing in mind that almost all the reanalysis datasets have significant inhomogeneities in the pre-satellite era, observation-based warming estimates should be treated carefully. However, as expected ERA-40 data show almost 1 K warming in the middle stratosphere before the eruption indicating downward propagation of warmer anomalies associated with the westerly QBO. Using radiosonde data, Free and Lanzante (2009) attributed around 1.5 K warming to the Mt. Agung volcanic aerosol near 50 hPa, which is somewhat consistent with ERA-40 (after removing 1 K warming due to the westerly QBO). However, almost all of the simulations show 1-2 K more warming compared to ERA-40 data as modelled temperature differences do not include QBO-related anomalies.

## 5   Conclusions

We have applied the interactive stratospheric aerosol configuration of the UM-UKCA model to simulate the formation and global dispersion of the volcanic aerosol clouds from the three largest tropical eruptions of the 20th century: Agung, El Chichón and Pinatubo. The simulations are analysed to assess the evolution of each eruption cloud, from an initial tropical reservoir of volcanic aerosol to a hemispherically dispersed stratospheric aerosol cloud. For each eruption, 3-member ensembles are carried out for each of upper, lower and mid-point of the literature range of $SO_2$ emission, aligning with the design of the co-ordinated HErSEA experiment, part of the multi-model ISA-MIP interactive stratospheric aerosol modelling initiative (see Timmreck et al., 2018). The analysis is also designed to provide new "microphysically-consistent and observationally-constrained" volcanic forcing datasets for climate models, to represent each eruption's surface cooling more realistically.

Simulated aerosol optical properties are compared against a range of satellite datasets. The model captures the observed variation in global stratospheric sulphur from 1991-1993 HIRS measurements very well, and experiments **Pin10** and **Pin14** defining a model-specific 10 to 14 Tg emissions uncertainty range and identifying a potential weighting to define a best-fit forcing dataset for Pinatubo. Our simulations also show that the aerosol decay rate is inversely pro-

portional to the SO$_2$ injection amount, illustrating how increased aerosol particle size causes faster sedimentation. The

model ensembles compare very well to mid-visible and near-infra-red aerosol extinction from SAGE-II measurements, although the model has sAOD high bias in the tropics, a common feature seen in interactive stratospheric aerosol models (e.g. Niemeier et al., 2009; Mills et al., 2016; Sukhodolov et al., 2018). We have also compared the Pinatubo ensembles to the three widely used forcing datasets (CMIP6–GloSSAC, Sato and Ammann) and we find that the **Pin14** model ensemble shows overall best agreement. A plateau in lower stratospheric tropical extinction seen in CMIP6-GloSSAC data

for almost one year after the Pinatubo eruption, is not reproduced in our simulations and thus remains as an open scientific question. The 10-14 Tg SO$_2$ emissions range for the model is lower than the 14-23 Tg observed to be present after the eruption (Guo et al., 2004b), and the tropical sAOD$_{550}$ high bias is consistent with the models missing an important removal process. Plausible suggestions for these are: a) the vertical redistribution of the volcanic cloud due to ash, b) changes in SO$_2$ oxidation due to OH decrease inside the plume, and c) too strong a subtropical barrier in the models.

The simulated R$_{eff}$ shows good agreement with CMIP6-GloSSAC data, although the model simulates a deeper global layer of enhanced SAD than in the 3$\lambda$ dataset (Luo, 2016). Simulated global-mean SW forcing (solar dimming) in run **Pin10** shows excellent agreement with the magnitude of the anomaly in the ERBE data, and the LW forcing in the model also matches well with the magnitude and shape of the ERBE anomaly. Assuming a 1 K colder temperature anomaly in ERA-Interim tropical temperatures due to the downward propagating QBO, a warming of 3 K near 50 hPa is well

simulated in both **Pin10** and **Pin14** simulations. Overall, most of the comparisons suggest that about 10-14 Tg SO$_2$ injection between 21-23 km is sufficient to simulate the climate and chemical impact of the Mt. Pinatubo eruption.

For the El Chichón eruption, there are significant differences between observation-based sAOD$_{550}$ estimates, hence evaluation of the simulations is somewhat restricted. However, NH mid-latitudes generally have a good quality observational data record, and sAOD$_{550}$ from run **Elc05** shows good agreement with CMIP6 and in the tropics model compares

better with Sato dataset. Our extinction comparisons also show that there are clear inhomogeneities in the CMIP6-GloSSAC data during this period, hence El Chichón-related aerosol properties must be treated with caution. Based on comparisons of the lower stratospheric warming of about 2 K, 5 Tg and 7 Tg SO$_2$ injections seem to be reasonable lower and upper limits for what is required to simulate observed temperature changes.

Finally, evaluation of Mt. Agung aerosol is more complicated due to much larger differences in the observation-based

datasets. Due to the westerly phase of QBO and timing of the eruption, CMIP6 data show a tropical peak in sAOD$_{550}$ within a month of the March eruption which is transported to SH mid-latitudes by October. The Sato dataset suggest two peaks in the tropics 8 and 14 months after the eruption. Run **Agu06** shows reasonable agreement with limited amount of observational extinction data, although that is not conclusive. Comparison with the lidar measurements from Lexington suggests that the UM-UKCA simulated vertical extent of the Agung cloud in the NH mid-latitudes is in best agreement for

**Agu09**. Comparisons with ERA-40 temperature anomalies also suggests that 3 K warming in the tropical stratosphere (2 K in the model simulation due to westerly phase of QBO). Assuming CMIP6-simulated sAOD$_{550}$ is realistic, 6 Tg and 9 Tg SO$_2$ injection seem to be the best lower and upper estimates required to simulate Mt. Agung-related aerosol in the UM-UKCA.

Overall, we have validated the interactive stratospheric aerosol configuration of the GA4 UM-UKCA model and have
shown the simulated aerosol properties for the Pinatubo ensemble are consistently in good agreement to a range of
satellite-based observational datasets. For Pinatubo, we have also compared to three different independent tests of
the radiative effects from the volcanic aerosol cloud: the ERBE flux anomaly timeseries in the SW and LW, and the
stratospheric warming in the ERA-interim reanalysis. These comparisons confirm that a 10 to 14 Tg emission flux of $SO_2$
would accurately represent the effects that the new forcing datasets would enact for Pinatubo in CCM integrations. For
El Chichón and Agung, the magnitude of the volcanic forcing is highly uncertain, the volcanic aerosol datasets used in
CMIP5 and CMIP6 historical integrations showing substantial differences. We contend there is substantial potential to
improve on this situation, by identifying consensus forcings from multi-model simulations (Timmreck et al., 2018), with
comparison to additional in-situ and active remote sensing measurements such as those being initiated within the SSiRC
data rescue activity (Antuna-Marrero et al., 2020; Mann et al., 2020).

*Data availability.*  Simulated aerosol data are publicly available from http://homepages.see.leeds.ac.uk/~fbsssdh/Dhomse2019_Volcanic_
Aerosol_Data/ We will get doi for Data once manuscript is online

.

*Author contributions.*  SD and GM led the initial experiments design, model simulations, data analysis and the writing of the paper. The
figures were prepared by SD. GM, SS and JCA retrieved and processed Lexington LIDAR data. GM, LM, LA and CJ contributed to-
wards this GA4-UM-UKCA interactive stratospheric chemistry-aerosol model capability. All co-authors contributed to either advising/co-
ordinating the UKCA developments, writing sections of the paper, performing evaluation and/or reviewing drafts of the paper.

*Competing interests.*  The authors declare that they have no conflict of interest.

*Acknowledgements.*  We thank Larry Thomason (NASA Langley) and Beiping Luo (ETH-Zurich) for useful discussions about use of
CMIP6-GloSSAC and CMIP6 datasets. We also acknowledge the contributions of James Brooke (Univ. Leeds), Nicolas Bellouin (Univ.
Reading), Anja Schmidt (Univ. Cambridge) and Mohit Dalvi (Joint Weather and Climate Research Programme, UK Met Office) in
helping to develop this interactive stratospheric chemistry-aerosol configuration of GA4-UM-UKCA. SD was supported by the NERC
SISLAC project (NE/R001782/1) and NCEO (NE/N018079/1). MPC thanks NCEO for funding (NE/R016518/1). SD, GM and KC re-
ceived funding via the NERC highlight topic consortium project SMURPHS ("Securing Multidisciplinary UndeRstanding and Prediction
of Hiatus and Surge periods", NERC grant NE/N006038/1). GM also received funding from the NCAS, via the ACSIS long-term sci-
ence programme on the Atlantic climate system. GM was also part-funded from the Copernicus Atmospheric Monitoring Service
(CAMS), one of six services that form Copernicus, the European Union's Earth Observation programme. Juan Carlos Antuna-Marrero

acknowledges travel and subsistence funding from CAMS that enabled his 1-month visit in March 2019 to the University of Leeds, with CAMS and SEE also co-funding Sarah Shallcross's PhD studentship. We thank the European Centre for Medium-Range Weather Forecasts for providing the ERA-interim meteorological reanalyses, which were obtained via the UK Centre for Environmental Data Access (CEDA). Simulations were performed on the UK ARCHER national supercomputing service and data analysis used the UK collaborative JASMIN data facility.

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

**Table 1.** Set up of UM-UKCA simulations.

| Simulation | Injection amount (Tg $SO_2$) | Date | Height (km) | QBO phase |
|---|---|---|---|---|
| **Pin00** | 0 | 15 June 1991 | NA | Easterly |
| **Pin10** | 10 | As **Pin00** | 21–23 | As **Pin00** |
| **Pin14** | 14 | As **Pin00** | As **Pin10** | As **Pin00** |
| **Pin20** | 20 | As **Pin00** | As **Pin10** | As **Pin00** |
| **Elc00** | 0 | 4 April 1982 | NA | Westerly |
| **Elc05** | 5 | As **Elc00** | 24–26 | As **Elc00** |
| **Elc07** | 7 | As **Elc00** | As **Elc05** | As **Elc00** |
| **Elc10** | 10 | As **Elc00** | As **Elc05** | As **Elc00** |
| **Agu00** | 0 | 17 March 1963 | NA | Westerly |
| **Agu06** | 6 | As **Agu00** | 20–22 | As **Agu00** |
| **Agu09** | 9 | As **Agu00** | As **Agu06** | As **Agu00** |
| **Agu12** | 12 | As **Agu00** | As **Agu06** | As **Agu00** |

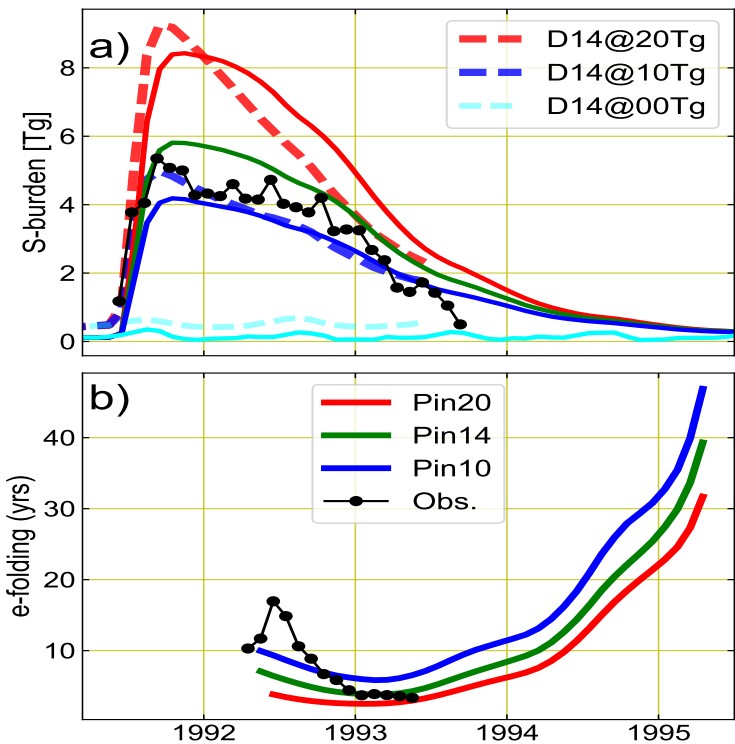

**Figure 1.** (a) Monthly mean stratospheric aerosol (globally integrated above 400 hPa) sulphur burden (S-burden) from simulations **Pin00** (aqua line), **Pin10** (blue line), **Pin14** (green line) and **Pin20** (red line). The S-burdens from Dhomse et al. (2014) for 0, 10 and 20 Tg SO$_2$ injection are shown with dashed aqua, blue and red lines, respectively. Estimated S-burden derived from High-resolution Infrared Radiation Sounder (HIRS) satellite measurements is shown with black dots (Baran and Foot, 1994). (b) S-burden decay rates (e-folding lifetime) calculated using simple linear fit using 7-month S-burden ($\pm 3$ for mid-point) time series.

Table 2: Some important aspects of the evaluation dataset.

| Aerosol property | Key Aspects |
|---|---|
| **Global stratospheric sulphur burden** | |
| High-resolution Infrared Radiation Sounder (HIRS) | 1.1 Derived from HIRS measurements onboard NOAA-10, -11, -12 satellites |
| | 1.2 Aqueous sulphuric acid aerosol retrieval using 8.2$\mu$m and 12.5 $\mu$m HIRS water vapour channels (Baran et al., 1993). |
| | 1.3 Derived sulphur burden based on assumed aerosol composition of 75% weight aqueous sulphuric acid solution droplets. |
| | 1.4 Global sulphur burden dataset is digitized from Figure 3 of Baran and Foot (1994) |

## Stratospheric AOD ($sAOD_{550}$) and extinction ($ext_{550}$) at 550 nm

| | |
|---|---|
| The CMIP6-GloSSAC forcing dataset https://eosweb.larc.nasa.gov/project/glossac/glossac) | 2.1 Pinatubo aerosol cloud primarily from SAGE-II, HALOE and CLAES observations (Thomason et al., 2018). |
| | 2.2 Improved Pinatubo gap-fill in mid-latitudes from combining SAGE-II with CLAES |
| | 2.3 Tropical Pinatubo gap-fill from combining SAGE-II with Mauna Loa lidar (SPARC, 2006). |
| | 2.4 El Chichon aerosol cloud mostly derived from high-latitude SAM-II data ($64°$N-$82°$N and $64°$S-$84°$S) |
| | 2.5 Tropical and mid-latitude El Chichon cloud from combining SAM-II with lidar data from 5 aircraft surveys ($13°$N to $80°$N) and from Hampton, Virginia ($37°$N). |
| The CMIP6-AER2D forcing dataset (ftp://iacftp.ethz.ch/pub_read/luo/CMIP6/) | 3.1 From 2D interactive stratospheric aerosol simulations (Arfeuille et al., 2014). |
| | 3.2 Primarily from the 8 major eruption clouds in 1850-1979 (29 in 1600-present dataset). |
| | 3.3 Additional minor eruption clouds from Stothers (1996) are also included. |
| The CMIP5-Sato forcing dataset (https://data.giss.nasa.gov/modelforce/strataer/) | 4.1 NASA GISS observation-based forcing data for 1850-2012 ($sAOD_{550}$ only). |
| | 4.2 Satellite era, uses SAGE-I, SAM-II, SAGE-II and OSIRIS measurements. |
| | 4.3 Pre-satellite era uses syntheses of different measurements |
| | 4.4 Surface radiation measurement dataset for Agung (Dyer and Hicks, 1968) highly uncertain in the tropics (Stothers, 2001). |
| The CMIP5-Ammann dataset (ftp://ftp.ncdc.noaa.gov/pub/data/paleo/climate_forcing/volcanic_aerosols/ammann2003b_volcanics.txt) | 5.1 Simple model-based dataset for 13 eruption clouds 1880-2000 ($sAOD_{550}$ only). |
| | 5.2 Based on parameterisation for meridional dispersion from tropical reservoir to mid-latitudes, determined by Brewer-Dobson circulation seasonal cycle. |
| | 5.3 12-month e-folding timescale for decay of tropical volcanic aerosol reservoir |
| | 5.4 Peak $sAOD_{550}$ for each eruption scaled to match aerosol loading from Stothers (1996); Hofmann and Rosen (1983b); Stenchikov et al. (1998), assuming $R_{eff}$ = 0.42 $\mu$m. |
| The post-Agung Lexington lidar dataset from Grams (1966) (see Supplementary Information) | 6.1 694nm backscatter ratio profiles from Lexington, Massachusetts ($42°$N, $71°$W Fiocco and Grams, 1964; Grams and Fiocco, 1967) ($ext_{550}$ only) |
| | 6.2 1-km dataset for 66 lidar soundings (Jan 1964 to July 1965) in Table A1 of Grams (1966). |
| | 6.3 Backscatter ratio timeseries at 15km, 20km and 24km tabulated into ASCII file. |
| | 6.4 Conversion to $ext_{550}$ using extinction-to-backscatter ratio from Jäger and Deshler (2003). |

## Vertical profile evolution of Effective Radius ($R_{eff}$) and Surface Area Density (SAD)

| | |
|---|---|
| CMIP6-GloSSAC (Pinatubo and El Chichon) and CMIP6-AER2D (Agung) | 7.1 SAD for Pinatubo and El Chichon aerosol clouds from GloSSAC, using SAGE-II 3-$\lambda$ method |
| | 7.2 SAD for Agung aerosol cloud from 2D interactive stratospheric aerosol model simulations |
| | 7.3 Volume concentration for each cloud derived from same method |
| | 7.4 Effective radius from 3 times ratio of volume concentration to SAD |

**Vertical profile of tropical stratospheric temperature anomaly**

From ECMWF reanalysis data

8.1  Temperature anomaly based on difference from 5-year mean starting in year of eruption

8.2  T-anomalies for Pinatubo and El Chichón from the ERA-interim reanalysis (Dee et al., 2011)

8.3  For the Agung period, anomaly derived from the ERA-40 year dataset (Uppala et al., 2005)

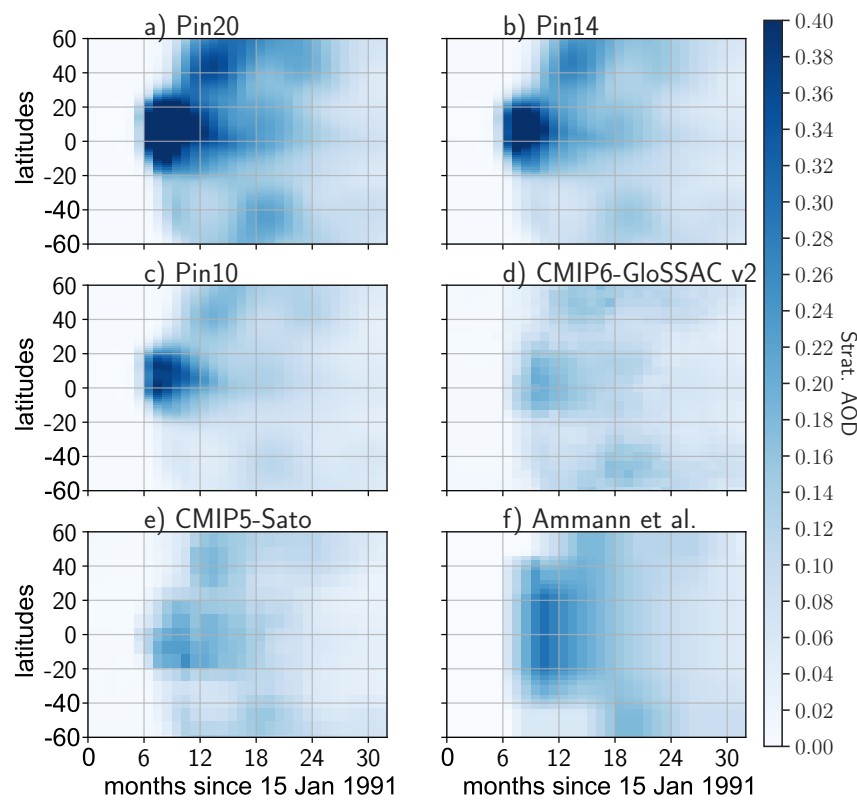

**Figure 2.** Ensemble mean stratospheric Aerosol Optical Depth (sAOD) from simulations (a) **Pin20**, (b) **Pin14**, and (c) **Pin10**. Panels (d)-(f) show sAOD$_{550}$ from CMIP6-GloSSAC (Thomason et al., 2018), Sato (Sato et al., 1993) and Ammann (Ammann et al., 2003), respectively.

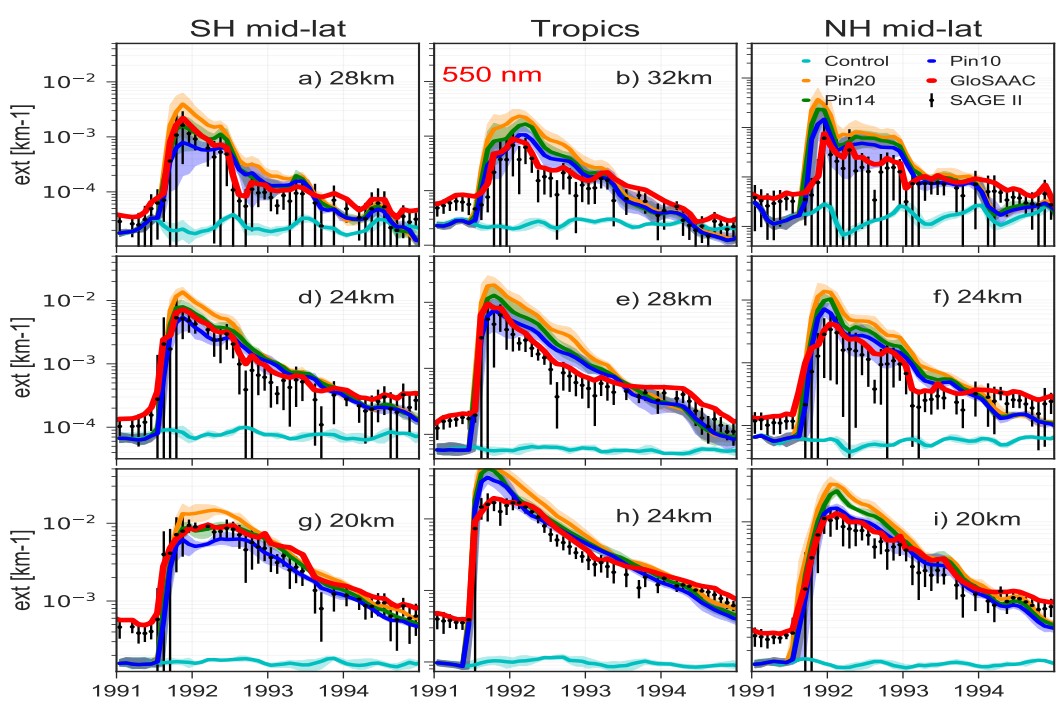

**Figure 3.** Ensemble mean extinctions (550 nm) from simulations **Pin00** (aqua), **Pin10** (blue), **Pin14** (green), and **Pin20** (orange). The shaded regions indicate the variability among ensemble members. Extinctions for SH mid-latitudes (35°S − 60°S (panels a, d, g)), tropics (20°S − 20°N (panels b, e, h )), and NH mid-latitudes (35°N − 60°N (panels c, f, i)) are shown in left, middle and right panels, respectively. Mid-latitude extinctions are shown for 20, 24 and 28 km, whereas tropical profiles are shown for 24, 28 and 32 km. Monthly mean extinction from SAGE II v7.2 measurements for a given latitude band are shown with black filled circles and vertical lines indicate standard deviation from all the measurements for a given month. Gap-filled extinctions from the CMIP6-GloSSAC v2 dataset (Kovilakam et al., 2020) are shown with a red line.

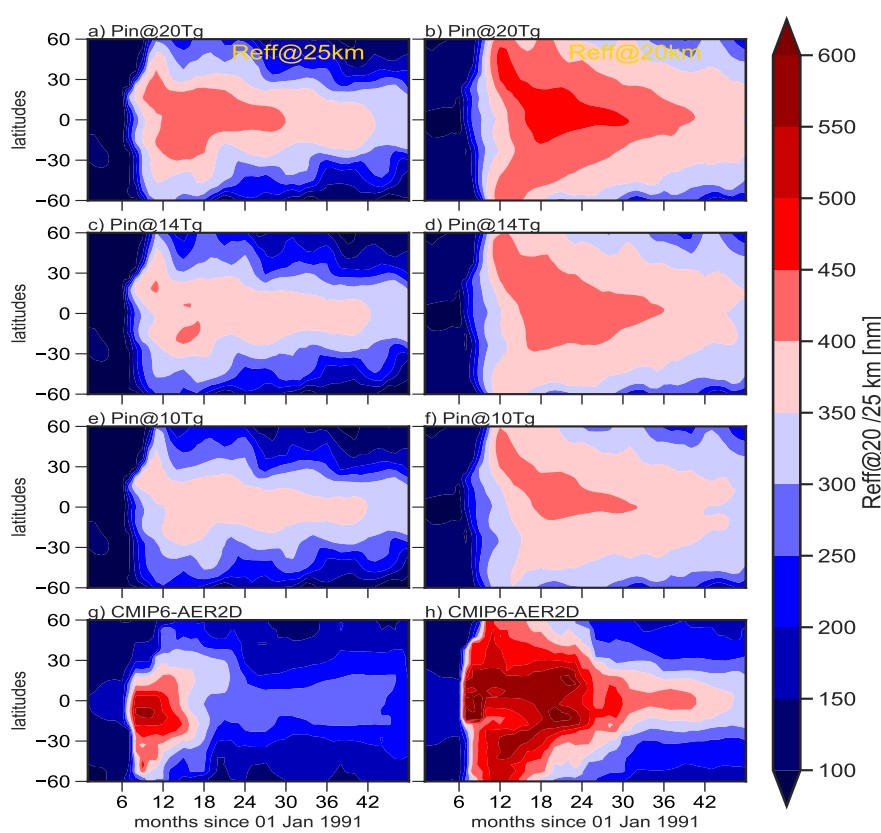

**Figure 4.** Modelled effective radii ($R_{eff}$, in $\mu$m) from (a, b) simulation **Pin20**, (c, d) simulation **Pin14**, (e, f) simulation **Pin10** and (g, h) CMIP6-GloSSAC V2 at (left) 25 km and (right) 20 km.

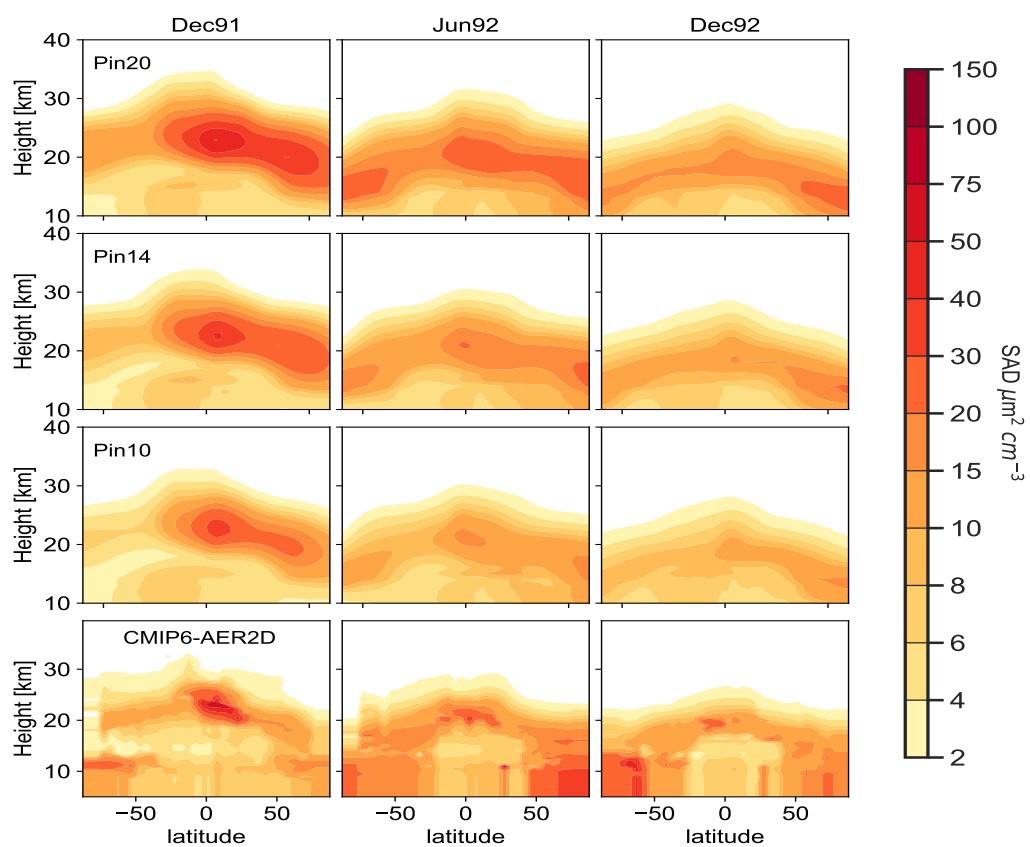

**Figure 5.** Zonal mean monthly mean Surface Area Density (SAD, $\mu m^2\ cm^{-3}$) for December 1991, June 1992 and December 1992 from ensemble mean simulations (top row) **Pin20**, (second row) **Pin14**, and (third row) **Pin10**. The bottom row shows observation-based SAD estimates from Arfeuille et al. (2014).

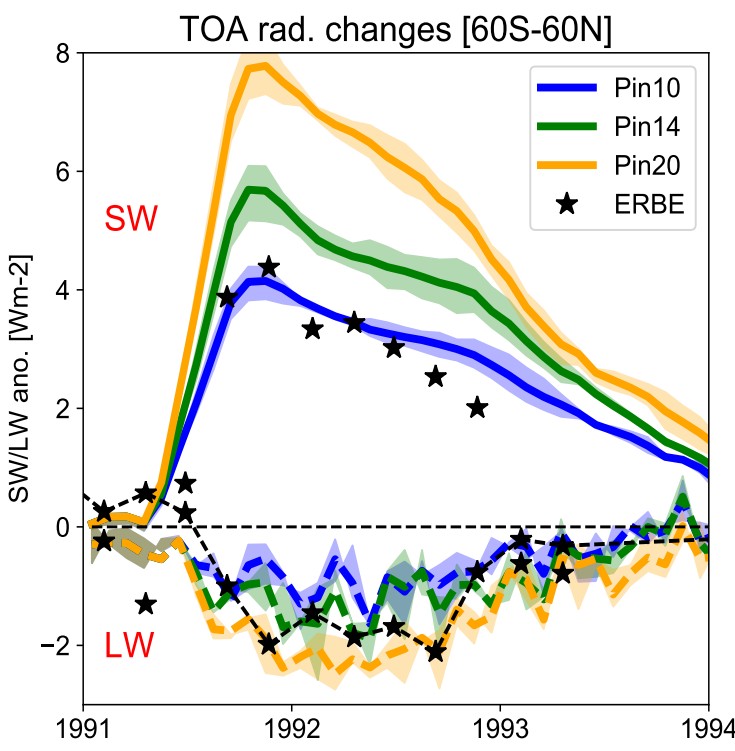

**Figure 6.** Near-global (60°S-60°N) longwave (LW) and shortwave (SW) heating anomalies (Wm$^{-2}$) from the ensemble mean of simulations **Pin20** (blue), **Pin14** (green), and **Pin10** (orange). Estimated anomalies from the Earth Radiation Budget Experiment (ERBE) satellite data are shown with black stars.

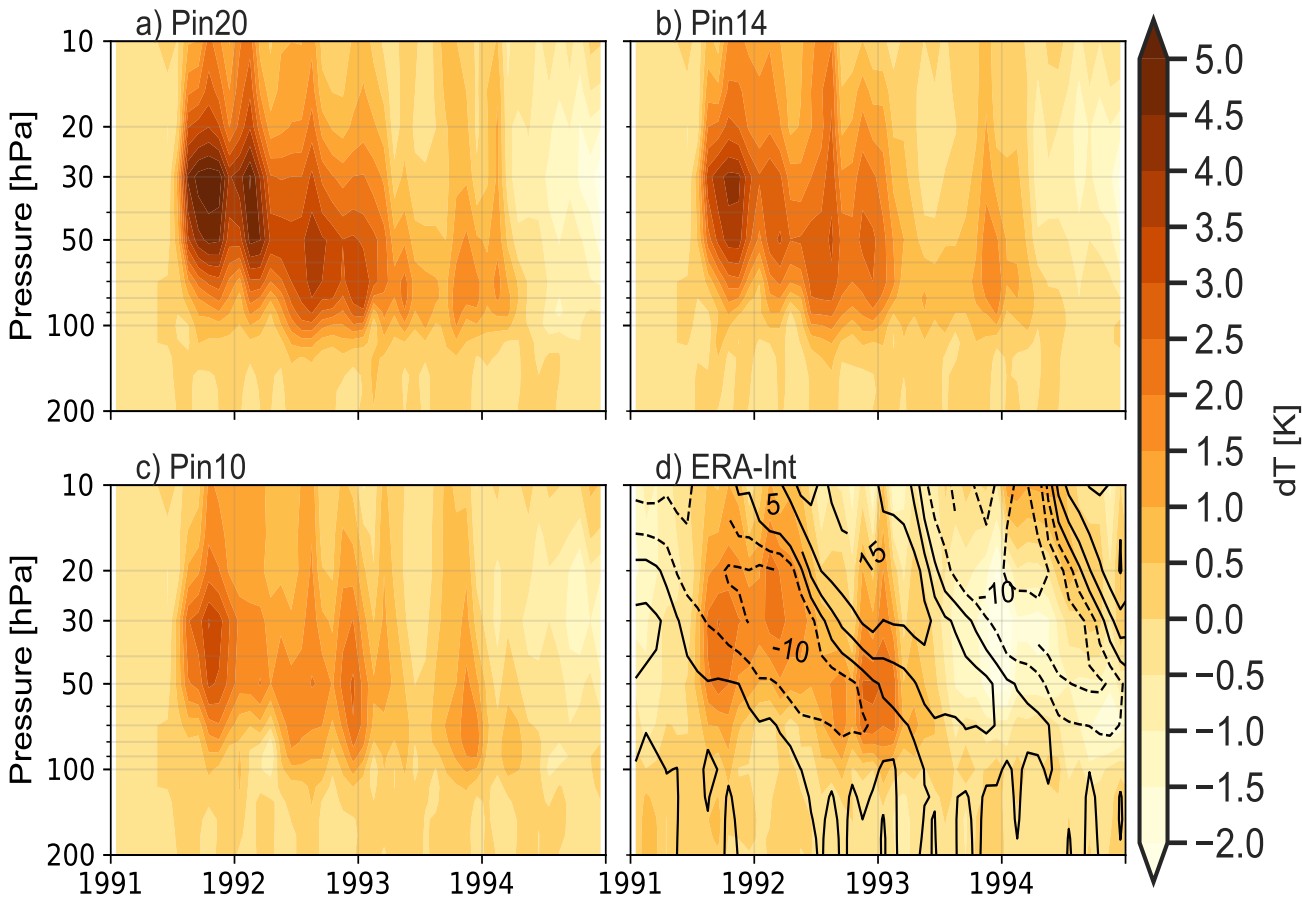

**Figure 7.** (a)-(c) Ensemble mean aerosol-induced heating (K) in the tropical (20°S – 20°N) stratosphere, calculated by subtracting temperature fields from a control simulation for simulations **Pin20**, **Pin14** and **Pin10**. (d) Tropical temperature (shaded) and zonal wind (contour) anomalies from ERA-Interim reanalysis data (for 1991–1995 time period). Contour intervals for wind anomalies are 4 m/s and negative anomalies are shown with dashed lines.

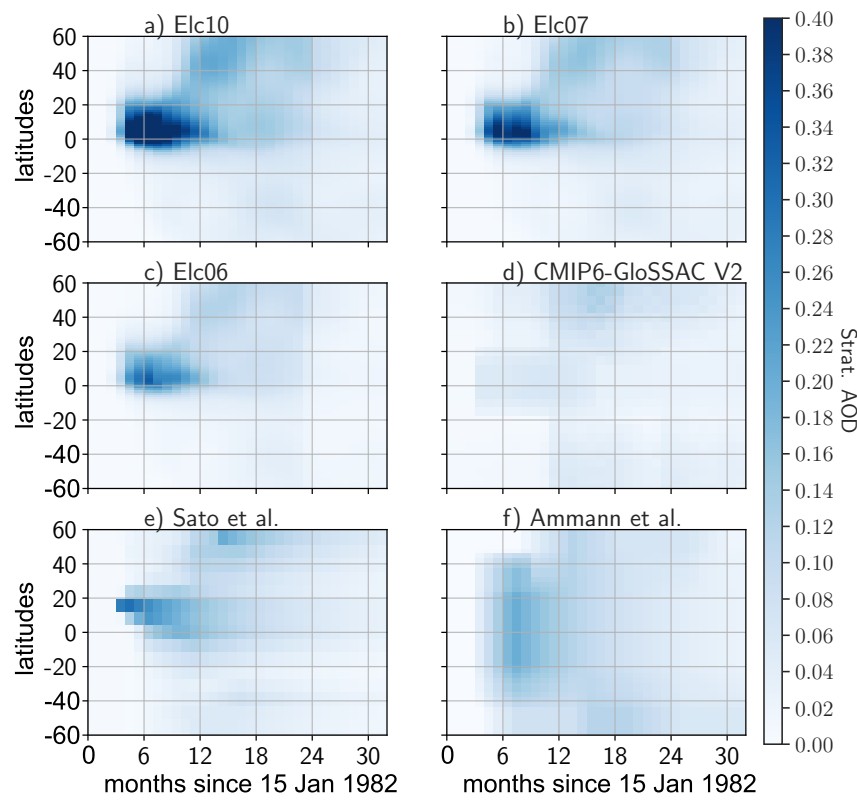

**Figure 8.** Same as **Fig.** 2, but for El Chichón simulations (a) **Elc10**, (b) **Elc07**, and (c) **Elc05**.

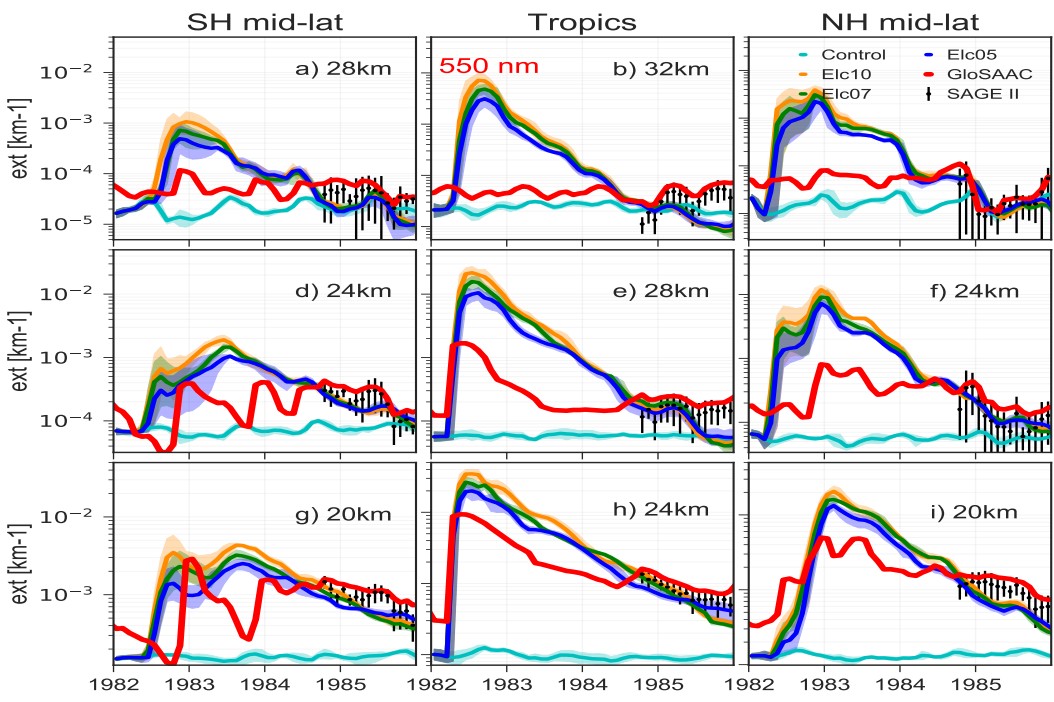

**Figure 9.** Same as **Fig.** 3, but for El Chichón simulations (a) **Elc05**, (b) **Elc07** and (c) **Elc10**.

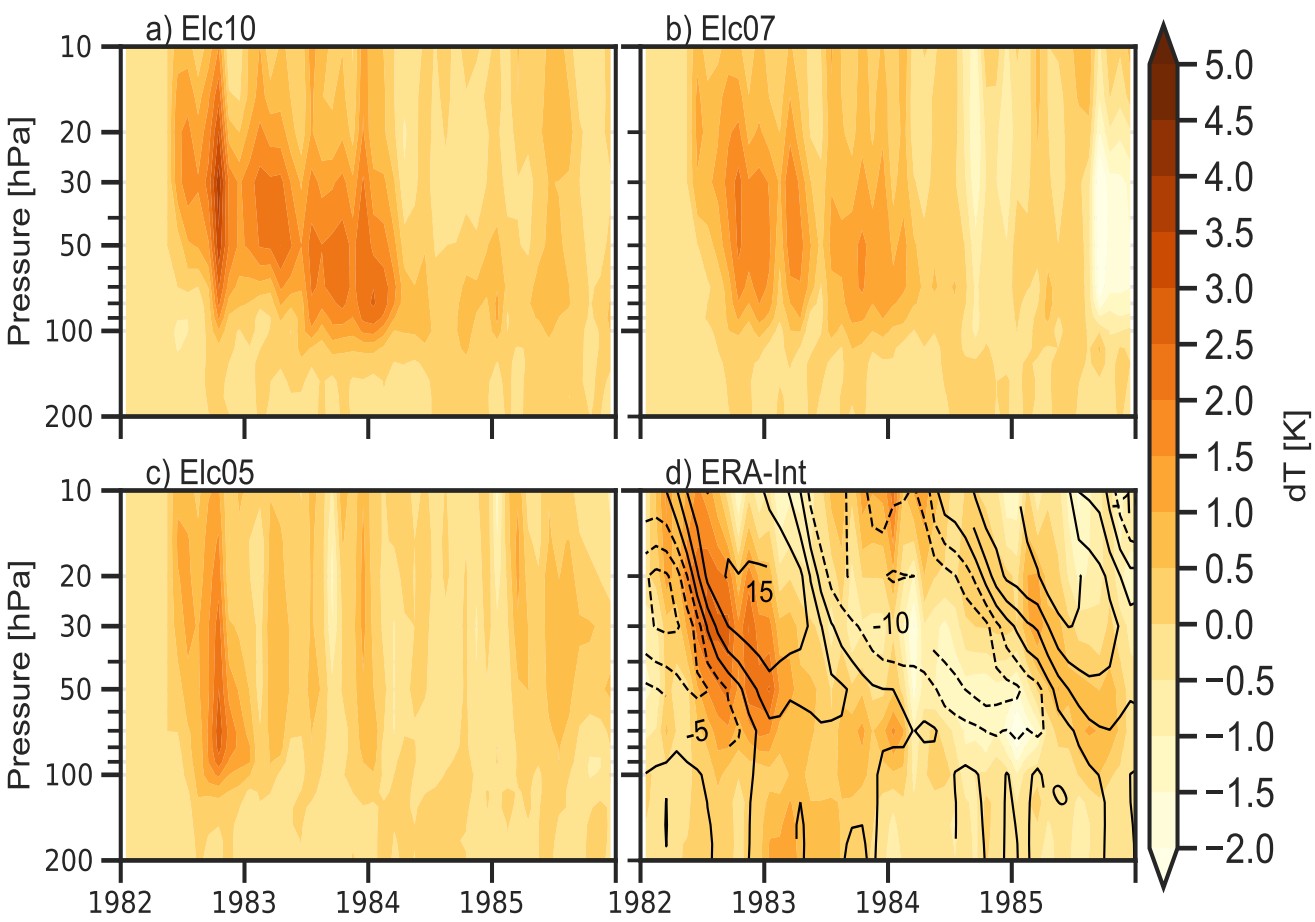

**Figure 10.** Same as **Fig.** 7, but for El Chichón simulations (a) **Elc10**, (b) **Elc07** and (c) **Elc05**.

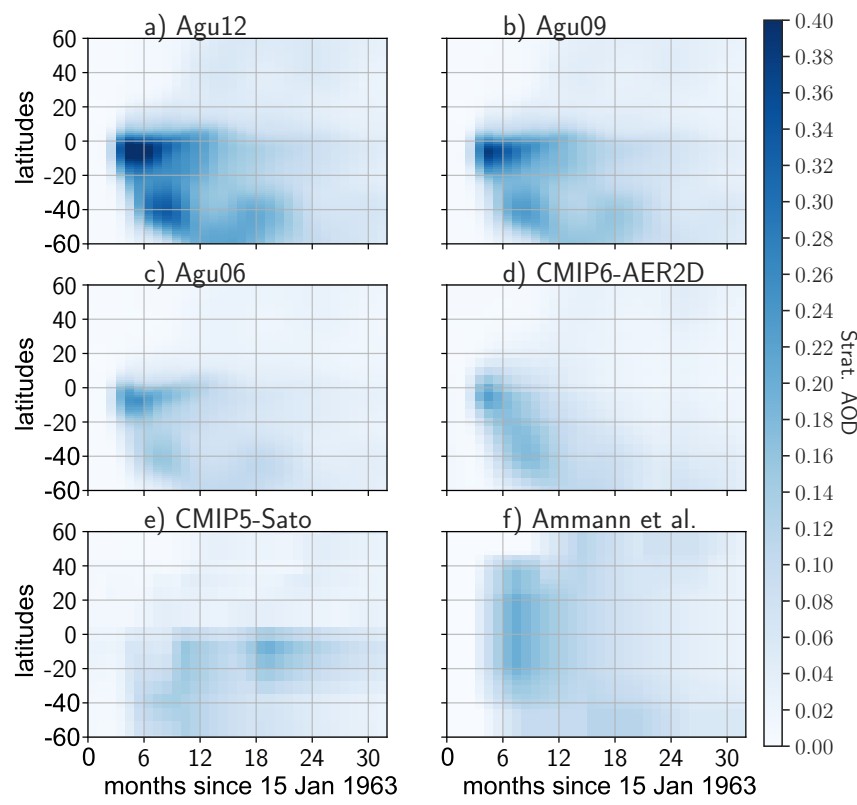

**Figure 11.** Same as **Fig.** 2, but for Mt. Agung simulations (a) **Agu12**, (b) **Agu09**, and (c) **Agu06**.

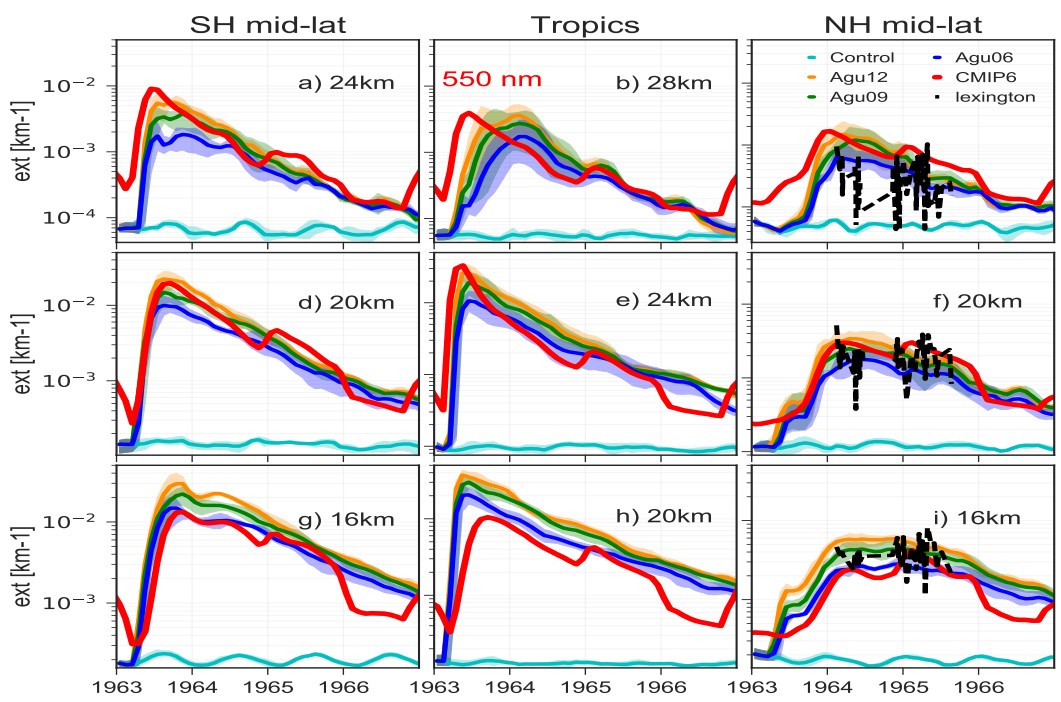

**Figure 12.** Same as **Fig.** 3, but for Mt. Agung simulations **Agu06**, **Agu09** and **Agu12**. Mid-visible aerosol extinction are shown at 16, 20 and 24 km for mid-latitudes and at 20, 24 and 28 km for the tropics. Also shown are aerosol extinction from the CMIP6-AER2D dataset (Arfeuille et al., 2014) and from LIDAR measurements at a NH mid-latitude site (Lexington, Massachusetts, USA) (Grams and Fiocco, 1967).

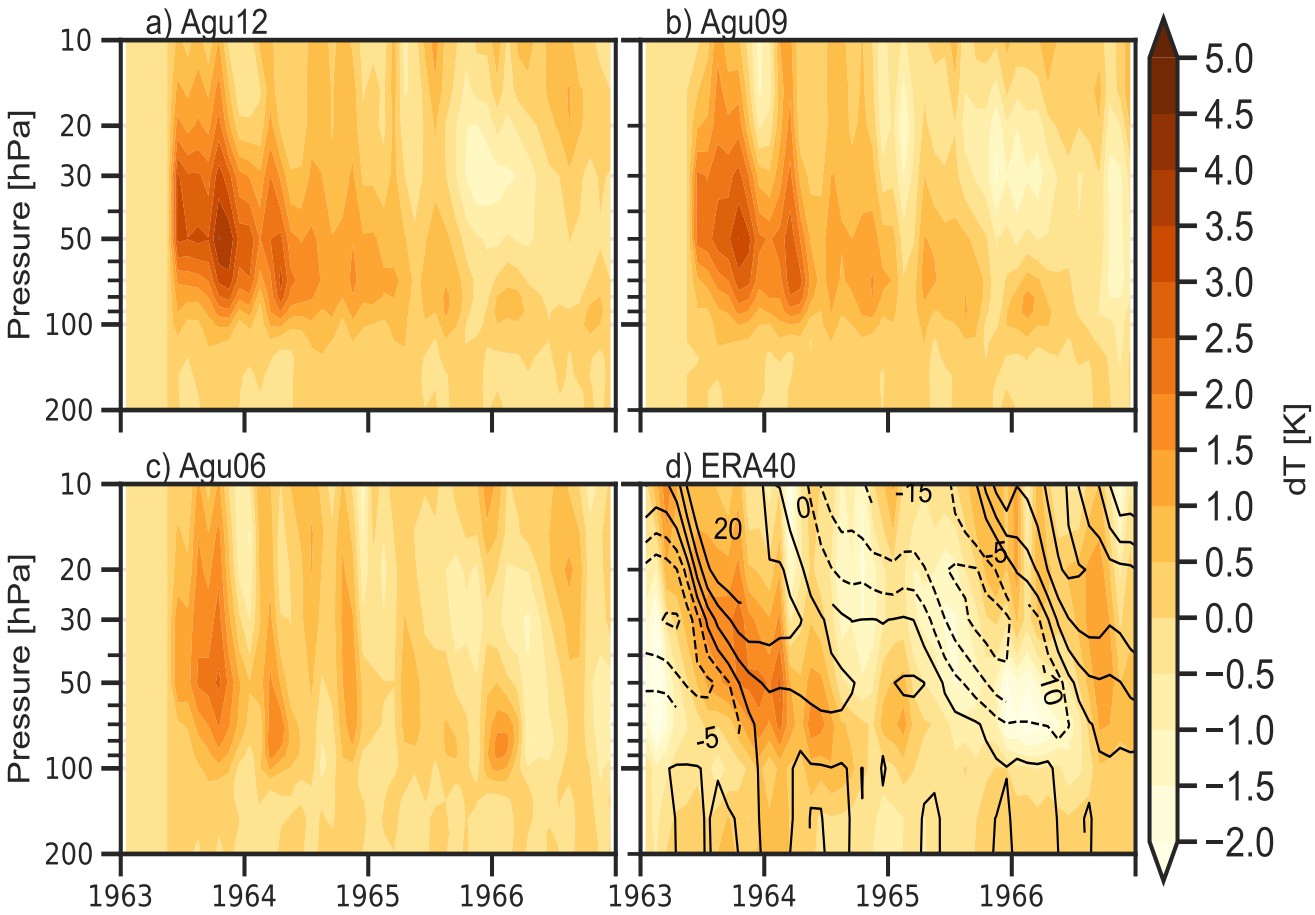

**Figure 13.** Same as **Fig.** 7, but for Agung simulations (a) **Agu12**, (b) **Agu09** and (c) **Agu06** and for (d) temperature anomalies calculated using ERA-40 reanalysis data.