# Peer review of "Evaluating the simulated radiative forcings, aerosol properties and stratospheric warmings from the 1963 Mt Agung, 1982 El Chichón and 1991 Mt Pinatubo volcanic aerosol clouds"

_Atmospheric Chemistry and Physics, 2020_

## Referee Comment (RC1) · Daniele Visioni (Referee) · 23 May 2020

Review of "Evaluating the simulated radiative forcings, aerosol properties and stratospheric warmings from the 1963 Agung, 1982 El Chichón and 1991 Mt Pinatubo volcanic aerosol clouds" by Dhomse et al.

This article gives an overview of the results from the UM-UKCA model simulations of the three biggest volcanic eruptions of the 20th century, and compares against available datasets. All simulations are run following the design of ISA-MIP.
In light of both CMIP6 and the release of the new generation of models, and also of ISA-MIP, of which this study is most likely the first showing results of the simulations described in Timmreck et al. (2018), I believe this study to be of great importance and a very good fit for ACP.

I have some suggestions to improve the presentation of the results and the discussion in this paper before it can be published. After these minor comments are addressed, the study can certainly be published in ACP.

Some broad comments:

• "Evaluation dataset" section: this section is a bit confused and hard to follow. I suggest a table for the supplementary (similar to Table 1), at least, that sums up all of this information, including columns for timespan, type of observation and link to the dataset.
• Supplementary: the reference is missing at line 4

In general, I suggest a more careful check of the grammar of the manuscript: some phrases seem to be written in haste, and it could make for a much more enjoyable read if the style was a bit easier to understand. I offer some examples below:

Lines 277-280: this phrase needs a bit of rewording, it's confusing.
Line 288: "the" lower end
Line 341: I think here you might be referring to the other Pitari et al. (2016) paper (Stratospheric Aerosols from Major Volcanic Eruptions: A Composition-Climate Model Study of the Aerosol Cloud Dispersal and *e*-folding Time) that discusses the effects of the QBO phase on the cloud dispersal.

Lines 343-345: While true that both cited paper mention the low altitude of the aerosols formed after the Hudson eruption, both remark that indeed the effect of that eruption was clearly distinguishable from the one from Pinatubo. From the conclusions of Pitts and Thomason (1993):

"Below 15 km, Cerro Hudson aerosols were transported poleward during September and remained a persistent feature beneath the vortex throughout the spring"

I understand that the experiments shown in this paper are part of ISA-MIP and thus part of a strict protocol, but I would just not be so quick in dismissing the Hudson eruption, especially in explaining the differences shown in Fig. 2 against the CMIP6 database, that are much larger in the southern hemisphere (where the Hudson eruption had more effect). I would like to see this discussed a little bit more in the manuscript (and, as a curiosity, see how the results change if this eruption is included, but I'm not suggesting to the authors do that for this work).

Line 371: "the more SO2 *is injected*"? and then, "within *the* first few months"
Line 375: *the* first three months

Line 378: *the* balance (there's a few of these here and there in the manuscript…)

Lines 385-388: this phrase needs to be checked, it seems like it's been written in haste and it doesn't make much sense

Lines 404-407: this phrase is also hard to follow: why is "and smaller" between parenthesis?

Line 462: Certainly, this phrase can be made a bit more coherent

Line 487: Why "enact" an sAOD? Seem like the wrong verb to use

Line 621: it is *a* common feature

Fig. 1: The legend is a bit hard to follow: there is no line for the (I assume) Pin00 simulation. The last phrase of the caption (…using simple linear fit using 6 month S-burden (±3) time series) should be expanded to better explain what the authors have done.

Fig. 2,8: The southern latitudes should either have a minus sign, or S after the number.

Fig. 3: Panel h) has some lines that go outside the frame. Also, some figures have Pin00 while some have "Control". More consistency would make it easier to understand.

Fig. 5: "lambda"

Fig. 12: Lexington with a capital L. Also, the dashed line is very hard to follow, since the noise seems to be high. Maybe having just the dots for the single observations is better?

Fig. 13: Labels are cut in this figure

**References**

Pitari, G., Genova, G. D., Mancini, E., Visioni, D., Gandolfi, I., & Cionni, I. (2016). Stratospheric aerosols from major volcanic eruptions: A composition-climate model study of the aerosol cloud dispersal and e-folding time. *Atmosphere*. https://doi.org/10.3390/atmos7060075

---

## Referee Comment (RC2) · Anonymous Referee #2 · 28 May 2020

This manuscript evaluates UM-UKCA simulations of the Agung, El Chichon, and Mt. Pinatubo eruptions and presents conclusions on the $SO_2$ injection amount that provides the best comparison with observations.

Overall, I think this is an interesting and well written manuscript. The evaluation is detailed and well presented, the graphs are mostly clear, and the discussion is well structured and relevant. The introduction is informative and gives a good overview of modeling and observational constraints.

[Figure]

I would see this manuscript more fitting to GMD, rather than ACP, but in any case I think this manuscript is publishable after minor changes.

**Comments**

- Line 60: it is correct that the stratospheric aerosol load was enhanced in both hemisphere , but one also needs to account for the Cerro Hudson eruption that increased the aerosols in the southern hemisphere shortly after the Pinatubo eruption. There is a comment about this later in the manuscript, but I think it would be useful to mention this here, too.

- Line 160: 10 years of spinup might not be enough for some slow adjusting variables such as age of air. Did the authors check that the stratosphere was indeed at equilibrium?

- Line 163: Three ensemble members is not many. Jones et al (2016, doi:10.1002/2016JD025001) showed that the dispersal is highly sensitive to the initial conditions. It would be useful to add, at least in the supplementary material, results from each of the ensemble members, to understand how the latitudinal dispersal varies within an ensemble.

- Line 163: The different injections all have the same altitude, but the injection altitude is also a degree of freedom. The chosen injection altitudes are all above the tropopause, but an larger injection with a lower boundary in the UTLS could deliver similar results. I understand that the setup of this experiment was dictated by SSiRC, but it would be interesting to comment on the importance of the vertical distribution of the injection.

- Line 165: Not sure what you mean with "for simplicity". For simplicity of set up or for simplicity of analyzing the results, as it reduces the degrees of freedom?

- Line 225: It is not really correct that satellite measurements constrain particle size. Some satellite instrument provide the Angstrom coefficient, which is a proxy (not a measurement) for size. The angstrom coefficient depends on the size of the particle but also on the composition of the particle and hydration.

- Line 250: Mann et al (2019b) and (2020) are conference abstracts. Does ACP allow them as references?

- Line 277: Larger injections produce a stronger upwelling (which push toward a longer S lifetime) and larger particle radii (which push toward shorter e folding time). Do your result imply that the net effect is driven by the particle size, rather than the changes in upwelling?

- Line 295: Is there a published paper or report that documents the changes brought up by increasing the resolution? Maybe something was published when the model with higher resolution was released?

- Line 306: How is stratospheric AOD calculated? Is aerosol extinction integrated above the tropopause or above a fixed altitude?

- Line 310: I'm confused by this. Are the authors referring to Fig.1, when they write that Pin20 best matches the satellite observed $SO_2$ estimates? Pin20 is the one that compare the worst with HIRS.

- Line 313: I am not sure it is fair to say that Pin10 has the best agreement. All of them, including Pin10, overestimate the peak sAOD in the tropics. 18 months after the eruption Pin20 seems actually to do better. A similar statement requires a metric such as the globally averaged root mean square error. I generally find the qualitative comparison a weak point of this manuscript. It is very difficult to judge which simulation is performing best just by looking at the figures.

- Line 344: Cerro Hudson is at 45S, 12 km could be above the tropopause.

- Line 390: I think Pin20 should be included in Fig. 4. Even if high biased, it is interesting to see how the effective radius scales with injection burden.

- Line 392: I am confused by this sentence. Pin20 is not shown, and between Pin14 and Pin10 I don't see any clear difference. There is a need of some kind of metric, such as mean error. Judging from the current plot, both simulations seems to perform pretty poorly when compared to the CMIP6 dataset (if that is a valuable benchmark)

- Line 405: I am not sure where to look to see this. Please specify latitude and months of the part of the plot that you are commenting on.

- Line 598: Figure 13, not 12, right? Also, take out either "as" or "hence"

**Minor comments and typos.**

- Line 3: aerofsol

- Line 7: here and in several other instances in the text the "2" of SO2 is not written as subscript

- Line 112: "to to"

- Line 182: closing parenthesis should not be there

- Line 269: "Applyig"

- Line 329: "disussed"

- Line 386: "model IS not resolving"

---

## Referee Comment (RC3) · Anonymous Referee #2 · 28 May 2020

I apologize, I forgot to include some minor comments on the figures:

- Figure 1: I think "blue line" should be "solid lines", otherwise I do not understand which lines I am supposed to look at.

- Figure 2: I find this kind of graphs (Fig 2, 5, 8, 7, etc) difficult to interpret. Next to the AOD, there should also be a panel with the absolute or relative difference between simulations and datasets. You could build a 3x3 table of graphs showing the difference between each of the three ensembles and each of the 3 datasets

[Figure]

- Figure 3: just to be clear, the variability among ensemble members is the ensemble spread, right? Min to max values per each month.

- Fig. 10: seeing the colors in panel d) is difficult, as lines become dense right where the warming happens. It would be better to make the lines light grey or change the color table to something with more diversity. Also, why not including the mean QBO also in the simulation graphs? Does the QBO changes between the experiments with and without eruptions?
* * *

---

## Author Comment (AC1) · 4 Aug 2020

**Replies are in blue colour and italics**

Review of "Evaluating the simulated radiative forcings, aerosol properties and stratospheric warmings from the 1963 Agung,1982 El Chichón and 1991 Mt Pinatubo volcanic aerosol clouds" by Sandip S. Dhomse et al.

**Review #1 by Daniel Visioni**

This article gives an overview of the results from the UM-UKCA model simulations of the three biggest volcanic eruptions of the 20[th] century, and compares against available datasets. All simulations are run following the design of ISA-MIP. In light of both CMIP6 and the release of the new generation of models, and also of ISA-MIP, of which this study is most likely the first showing results of the simulations described in Timmreck et al. (2018), I believe this study to be of great importance and a very good fit for ACP. I have some suggestions to improve the presentation of the results and the discussion in this paper before it can be published. After these minor comments are addressed, the study can certainly be published in ACP.

*--> We thank Dr Visioni for these positive comments.*

**Some broad comments:**

"Evaluation dataset" section: this section is a bit confused and hard to follow. I suggest a table for the supplementary (similar to Table 1), at least, that sums up all of this information, including columns for timespan, type of observation and link to the dataset.

*--> Thank you for a very useful suggestion. We decided to add the suggested table into the main article (new Table 2) rather than the supplementary, so as to provide a summary of the important details for each observation dataset (wavelengths, data source) and to reference the papers and/or web-links to the individual datasets.*

*We have pasted below the Table 2 added to the dataset, the process also alerting us to correct some aspects of the text of the evaluation datasets section (see track-changes manuscript). For example, we improved the text re: the Pinatubo period in GloSSAC to read:*

```
"For the Pinatubo period, GloSSAC is an updated version of the gap-
filled dataset described in \citet[][chapter 4]{SPARC2006},
combining SAGE II aerosol extinction (in the solar part of the
spectrum), with HALOE and CLAES aerosol extinction in the infra-red
\citep[see][]{Thomason2018}. For the period where the SAGE-II signal
was saturated (e.g. Thomason, 1992), GloSSAC applies an improved
gap-fill method in mid-latitudes, but in the tropics is still based
on the composite dataset from \citet[][pages 140-147]{SPARC2006},
combining with ground-based lidar measurements from Mauna Loa,
Hawaii (19.5°N, \citep{Barnes1997}), and after January 1992 also
with lidar measurements from Camaguey, Cuba \citep[23°N,
see][]{Antuna1996})."
```

Table 2: Some important aspects of the evaluation dataset.

[revised manuscript text omitted]

**Minor comments: Careful rewording**

a) Supplementary: the reference is missing at line 4. In general, I suggest a more careful check of the grammar of the manuscript: some phrases seem to be written in haste, and it could make for a much more enjoyable read if the style was a bit easier to understand. I offer some examples below:

b) Lines 277-280: this phrase needs a bit of rewording, it's confusing.

c) This again confirms that the more SO2 injection leads to the faster particle growth, hence quicker removal within first few months after the eruption.

d) Line 288: "the" lower end.

*e)* Line 341: I think here you might be referring to the other Pitari et al. (2016) paper (Stratospheric Aerosols from Major Volcanic Eruptions: A Composition-Climate Model Study of the Aerosol Cloud Dispersal and e-folding Time) that discusses the effects of the QBO phase on the cloud dispersal.

*--> We agree with the reviewer. Some of the sentences were confusing and had some grammatical errors. We apologise for this. We have had a careful read and worked on the flow of the manuscript. The reference has been corrected.*

Lines 343-345: While true that both cited papers mention the low altitude of the aerosols formed after the Hudson eruption, both remark that indeed the effect of that eruption was clearly distinguishable from the one from Pinatubo. From the conclusions of Pitts and Thomason (1993): "Below 15 km, Cerro Hudson aerosols were transported poleward during September and remained a persistent feature beneath the vortex throughout the spring" I understand that the experiments shown in this paper are part of ISA-MIP and thus part of a strict protocol, but I would just not be so quick in dismissing the Hudson eruption, especially in explaining the differences shown in Fig. 2 against the CMIP6 database, that are much larger in the southern hemisphere (where the Hudson eruption had more effect). I would like to see this discussed a little bit more in the manuscript (and, as a curiosity, see how the results change if this eruption is included, but I'm not suggesting to the authors do that for this work).

*--> We agree with the reviewer that we cannot dismiss the influence of Mt. Hudson eruption, and our wording was somewhat dismissive, hence we have reworded the sentence. As GloSSAC V2 data became available, we have updated Figure 2 and, as reviewer pointed out, differences are indeed significant. Hence, we expanded discussion about Mt. Hudson eruption.*

Line 371: "the more SO2 is injected"? and then, "within the first few months"

*--> We reworded it as:*

*"This again confirms that the more SO2 injection leads to the faster particle growth, hence quicker removal within the first few months after the eruption."*

Line 375: the first three months

*--> Done.*

**Additional comments added after the original upload of the above reply to reviewers**

*Shortly after uploading our replies to the reviewers (including AC1 above), and when finalising the revised manuscript, we discovered two subtle but important mistakes in the Python code used to generate the figures in the ACP-Discussions manuscript:*

1) *Figure 4: Typo in the code used to calculate effective radius: assigned the accumulation-soluble mode $H_2SO_4$ mmr to accumulation-insoluble $H_2SO_4$ mmr.*

   *A subtle typo in the code used to calculate effective radius caused an error in the initial assignment of modal $H_2SO_4$ component mass mixing ratios. The typo caused the calculation of total particle volume (PVOL) to double-count the accumulation-soluble mode $H_2SO_4$ mass mixing ratios (mmr), the accumulation-insoluble $H_2SO_4$ mode mmr not used in the calculations as a consequence. Specifically, the excerpt of code:*

```
H2SO4_nucsol_mmr=nc_fid.variables['NUCLEATION_MODE__SOLUBLE__H2SO4_MMR'][:]
H2SO4_Aitsol_mmr=nc_fid.variables['AITKEN_MODE__SOLUBLE__H2SO4_MMR'][:]
H2SO4_accsol_mmr=nc_fid.variables['ACCUMULATION_MODE__SOL__H2SO4_MMR'][:]
H2SO4_corsol_mmr=nc_fid.variables['COARSE_MODE__SOLUBLE__H2SO4_MMR'][:]
H2SO4_accins_mmr=nc_fid.variables['ACCUMULATION_MODE__SOL__H2SO4_MMR'][:]
```

   *should have been:*

```
H2SO4_nucsol_mmr=nc_fid.variables['NUCLEATION_MODE__SOLUBLE__H2SO4_MMR'][:]
H2SO4_Aitsol_mmr=nc_fid.variables['AITKEN_MODE__SOLUBLE__H2SO4_MMR'][:]
H2SO4_accsol_mmr=nc_fid.variables['ACCUMULATION_MODE__SOL__H2SO4_MMR'][:]
H2SO4_corsol_mmr=nc_fid.variables['COARSE_MODE__SOLUBLE__H2SO4_MMR'][:]
H2SO4_accins_mmr=nc_fid.variables['ACCUMULATION_MODE__INS__H2SO4_MMR'][:]
```

   *For the major volcanic aerosol cloud simulations analysed here, the majority of sulphuric acid mass is in that double-counted accumulation-soluble mode.*

   *Hence the typo caused the particle volume PVOL, used in the calculation of the model's effective radius (=3\*PVOL/SAREA) to be much higher.*

   *This affected the effective radius values shown in Figures 4 a), b), d), e) in the ACPD article to be much higher than their true values.*

*2) Figures 2d), 8d),11d): Error in sAOD calculated for CMIP6 dataset (depth error).*

*The stratospheric AOD values shown for the CMIP6 representations of the Pinatubo, El Chichon and Agung aerosol in Figures 2d, 8d, 11d of the ACPD article are factor 2 too high, due to an error in the depth used in the calculations.*

*The depth error arises within our code to integrate to sAOD the altitude-resolved aerosol extinction dataset provided with the CMIP6 volcanic aerosol dataset. When calculating the sum over vertical levels, the depth assigned when integrating the aerosol extinction to stratospheric Aerosol Optical Depth (sAOD) was set at 1.0 km rather than 0.5 km, calculated sAOD was then a factor of 2 too high.*

*We apologise for both bugs. Whereas the error 1) was more subtle, and the Reff from the model being too high was not obvious, we should have realised that the CMIP6 sAOD values shown in those figures were a factor-2 too high. Even though there is no documentation paper for the pre-satellite part of the CMIP6 dataset (CMIP6-AER2D) sAOD, we should still have realised this error when preparing the manuscript. We are relieved to have found this error during the review process and in the revised manuscript, the Figures 2d, 8d and 11d show the correct sAOD$_{525}$ values.*

*The typo explained in 1) is now remedied, and the simulated Reff values shown in Figures 4c) to f) of the revised manuscript represent the model predictions correctly.*

*We also added the two extra panels requested by Reviewer 2 to additionally show the 20Tg simulation Reff field at 25km (Figure 4a) and 20km (Figure 4b).*

*Note that the Reff figure in the Supplementary Material (Figure S6) has also been updated since the ACPD article to show the correct values. There only the 10Tg and 20Tg runs are shown to match the runs used in Dhomse et al. (2014), enabling comparison with the corresponding figure in that paper.*

*With these changes to the simulated Reff values in Figure 4 and Figure S6, the Section 4.1 text analysing the Reff variations (lines 389-419 in the ACPD article) has been re-written to:*

"Next, we evaluate the meridional, vertical and temporal variations
in effective radius (Reff) in the Pinatubo UM-UKCA datasets. The
particle size variations in these interactive simulations of the
Pinatubo cloud reflect the chemical and microphysical processes
resolved by the chemistry-aerosol module, in association with the
stratospheric circulation and dynamics occurring in the general
circulation model. We analyse these model-predicted size variations
also alongside those in the benchmark observation-based Reff dataset
from CMIP6-GloSSAC, which applies the 3-lamda size retrieval from

the 453nm, 525nm and 1020nm aerosol extinction measurements from SAGE-II (Thomason et al., 1997a, 2018).

Figure 4 shows zonal mean Reff at 25km, within the altitude range of the volcanic $SO_2$ injection, and at 20km, underneath the main volcanic cloud, results shown from 3-member means from the 10, 14 and 20Tg $SO_2$ emission runs (Pin10, Pin14 and Pin20). For comparability with the equivalent Figure from Dhomse et al. (2014), we also show in the Supplementary Material (Figure S6) the updated comparison to the Bauman et al. (2003) Reff dataset, for the corresponding Pin10 and Pin20 runs. Overall, the model captures the general spatio-temporal progression in the Reff variations seen in the GloSSAC dataset. However, whereas the 10Tg and 14Tg simulations agree best with the HIRS-2 sulphur-burden (Figure 1) and the GloSSAC sAOD and extinction (Figures 2 and 3), the magnitude of the Reff enhancement is best captured in the 20Tg run (Pin20). The comparisons suggest the low bias in simulated Reff seen in the previous UM-UKCA Pinatubo study (Dhomse et al., 2014) continues to be the case here. However, this low-bias in particle size/growth may simply be reflecting the required downward-adjustment of the Pinatubo $SO_2$ emission, a larger Reff enhancement in the 20Tg simulation clearly apparent. It is possible that the two-moment modal aerosol dynamics in GLOMAP-mode may affect its predicted Reff enhancement. However, the model requirement for reduced $SO_2$ emission is attributed to likely be due to a missing, or poorly resolved, model loss pathway, such as accommodation onto co-emitted volcanic ash. The sustained presence of ash within the Pinatubo cloud (e.g. Winker and Osborne, 1992) will likely have altered particle size and growth rates in the initial months after the eruption.

In the tropics, where Reff increases are largest, the timeseries of Reff is noticeably different in the core of the tropical reservoir (10°S to 10°N) to that in the edge regions (10°N-20°N and 10°S-20°S), at both 20km and 25km. The Reff increases in these edge regions occur when tropics to mid-latitude transport is strongest, in phase with the seasonal cycle of the Brewer-Dobson circulation, which tends to transport air towards the winter pole (Butchart, 2014). The Reff increases are due primarily to particle growth from coagulation and condensation, and the simulations also illustrate how the simulated Pinatubo cloud comprises much smaller particles at 25km than at 20km. The 25km level is in the central part of the Pinatubo cloud, particles there being younger (and smaller), because the oxidation of emitted volcanic $SO_2$ that occurs at that level, triggers extensive new particle formation in the initial months after the eruption (e.g. Dhomse et al., 2014). By contrast, at the 20km level, particles there will almost exclusively have sedimented from the main cloud, and therefore be larger. There is a slow but sustained increase in average particle size in the equatorial core of the tropical Pinatubo cloud, with the 20km level reaching peak Reff values only during mid-1992, in contrast to the peak S-burden and $sAOD_{550}$, which have already peaked at this time, being in decay phase since the start of 1992 (see Figures 1 and 2).

Whereas the simulated peak Reff enhancement occurs by mid-1992 in the tropics, the peak Reff in NH mid-latitudes occurs at the time of

peak meridional transport, the Reff variation there reflecting the seasonal cycle of the Brewer-Dobson circulation, as also seen in the tropical reservoir edge region.  The different timing of the volcanic Reff enhancement in the tropics and mid-latitudes is important when interpreting or interpolating the in-situ measurement record from the post-Pinatubo OPC soundings from Laramie (Deshler et al., 2003). Russell et al. (1996) show the Reff values derived from Mauna Loa ground-based remote sensing are substantially larger than those from the dust-sonde measurements at Laramie. The interactive Pinatubo simulation here confirm this expected meridional gradient in effective radius, with the chemical, dynamical and microphysical processes also causing a vertical gradient in the tropical to mid-latitude Reff ratio. The current ISA-MIP activity (Timmreck et al., 2018) brings a potential opportunity to identify a consensus among interactive stratospheric aerosol models for the expected broad-scale spatio-temporal variations in uncertain volcanic aerosol metrics such as effective radius."

*With the 1km-depth error in the integration of the CMIP6 aerosol extinction, and the subsequent correction to the sAOD shown for CMIP6-GloSSAC/CMIP6-AER2D in Figures 2d, 8d and 11d, there have also been some minor changes to interpret the evaluation of the UM-UKCA volcanic simulations. The revisions here are only minor changes in emphasis re: the comparisons to GloSSAC sAOD, and since the text is mainly analysing the sAOD variation, these changes are only minor.*

*The text changes for this CMIP6 sAOD correction are the lines 371-415, 562-583 and 623-631 of the revised (lines 446-491, 672-694 and 735-755 in a tracked change version) manuscript, corresponding to lines 316-357, 484-504 and 542-563 of the ACP Discussions article*

---

## Author Comment (AC2) · 4 Aug 2020

**Replies are in blue colour and italics**

Review of "Evaluating the simulated radiative forcings, aerosol properties and stratospheric warmings from the 1963 Agung,1982 El Chichón and 1991 Mt Pinatubo volcanic aerosol clouds" by Sandip S. Dhomse et al.

**Anonymous Reviewer #2**

This manuscript evaluates UM-UKCA simulations of the Agung, El Chichon, and Mt. Pinatubo eruptions and presents conclusions on the SO2 injection amount that provides the best comparison with observations. Overall, I think this is an interesting and well written manuscript. The evaluation is detailed and well presented, the graphs are mostly clear, and the discussion is well structured and relevant. The introduction is informative and gives a good overview of modeling and observational constraints. I would see this manuscript more fitting to GMD, rather than ACP, but in any case I think this manuscript is publishable after minor changes.

**Comments**

Line 60: it is correct that the stratospheric aerosol load was enhanced in both hemispheres, but one also needs to account for the Cerro Hudson eruption that increased the aerosols in the southern hemisphere shortly after the Pinatubo eruption. There is a comment about this later in the manuscript, but I think it would be useful to mention this here, too.

--> *We prefer to keep this paragraph as an overview of the main differences between the 3 eruption clouds. As also suggested by Reviewer #1, we modified the discussion to mention that the Cerro Hudson aerosol cloud may have contributed to the model - GloSSAC sAOD differences in the Southern Hemisphere. However, as we explain in the manuscript, SAGE-II measurements (Pitts and Thomason, 1993) and lidar measurements from Aspendale, Australia (Barton et al., 1992) clearly show that the Hudson aerosol was only a minor contributor to the total optical depth over the two volcanic aerosol clouds. We have instead modified the text to say:*

```
"One thing to note is that our simulations do not include the source
of volcanic aerosol from the August 1991 Cerro Hudson eruption in
Chile. However, measurements from SAGE II \citep{Pitts1993} and
ground-based lidar \citep{Barton1992} indicate that the Hudson
aerosol cloud only reached to around 12 km, with the Pinatubo cloud
by far the dominant contributor to SH mid-latitude sAOD. So,
although we have not included the Mt. Hudson aerosol in our
simulations, we argue this was only a minor contributor to the
differences between model and GloSSAC V2 sAOD, and does not explain
why 20Tg SO2 injection (\pind~) shows best agreement in the SH."
```

Line 160: 10 years of spin-up might not be enough for some slow adjusting variables such as age of air. Did the authors check that the stratosphere was indeed at equilibrium?

*Yes, we analysed age of air and long-lived tracers in each 20-year timeslice run to check that the model was fully spun up. In the revised manuscript we added a sentence to explain the exact procedure we followed, that paragraph now reading:*

```
"For each 20-year time-slice run, we analysed the stratospheric
sulphur burden, ozone layer and the distributions of age of air and
selected long-lived tracers, to check that the model had fully
adjusted to the GHG and ODS setting. We then analysed timeseries of
the tropical zonal wind profile, to then identify three different
model years that gave QBO transition approximately matching that
seen in the ERA-interim re-analysis \citep{Dee2011}, the
initialisation fields for those years then used to re-start the
three ensemble member transient runs. We show the QBO evolution for
each Pinatubo simulation in the Supplementary Material (Figure S1)."
```

Line 163: Three ensemble members is not many. Jones et al (2016, doi:10.1002/2016JD025001) showed that the dispersal is highly sensitive to the initial conditions. It would be useful to add, at least in the supplementary material, results from each of the ensemble members, to understand how the latitudinal dispersal varies within an ensemble.

*--> We thank the reviewer for pointing this out. We agree that some readers might be interested in this comparison. We note that the SO₂ injection altitude in Jones et al., (2016) is closer to the tropopause than in our simulations, which we think may explain why they observe such large differences in sAOD evolution. We also note that the interactive stratospheric aerosol simulations in Jones et al. (2016) used a simpler (single-moment) aerosol scheme, so sulphate aerosol particles form immediately at the assumed size as the SO₂ oxidises. By contrast, in our aerosol microphysics module, the particles grow from initial nanometre sizes according to the timescales of the microphysical processes (coagulation and condensation). That particles form immediately at radiation-interacting sizes might also have contributed to the larger variation between ensemble members in Jones et al. (2016). We have added to the Supplementary Material an extra figure showing the sAOD evolution for each ensemble member.*

Line 163: The different injections all have the same altitude, but the injection altitude is also a degree of freedom. The chosen injection altitudes are all above the tropopause, but a larger injection with a lower boundary in the UTLS could deliver similar results. I understand that the setup of this experiment was dictated by SSiRC, but it would be interesting to comment on the importance of the vertical distribution of the injection.

*--> We agree that interactive stratospheric aerosol simulations of volcanic aerosol clouds are sensitive to the assumed injection height, and this is part of the rationale for the HErSEA experiments within ISA-MIP. This sensitivity to injection height is discussed extensively in Timmreck et al., (2018) and Marshall et al., (2018) (https://doi.org/10.1029/2018JD028675).*

*The reviewers is incorrect in stating that setup of the experiment was dictated by SSiRC, the rationale for the experiment is explained clearly in the HErSEA section of the Timmreck et al. (2018) paper. The HErSEA design involves three alternative injection height simulations, which we have carried with UM-UKCA, but here we only show one of the three injection height "eruption realisations" included in the HErSEA design.*

Line 165: Not sure what you mean with "for simplicity". For simplicity of set up or for simplicity of analyzing the results, as it reduces the degrees of freedom?

*--> Good point. Indeed simplicity was referring to terms of model setup as well as our attempt to avoid any complicated ozone chemistry feedback mechanism. However, to avoid the confusion we have deleted "For simplicity" as our approach is focussed on the aerosol evolution.*

Line 225: It is not really correct that satellite measurements constrain particle size. Some satellite instrument provide the Angstrom coefficient, which is a proxy (not a measurement) for size. The Angstrom coefficient depends on the size of the particle but also on the composition of the particle and hydration.

*--> Whilst we agree that in-situ measurements are the primary ground-truth for evaluating stratospheric aerosol particle size distribution, the multi-wavelength algorithm for particle surface area density and particle volume concentration used in the GloSSAC-derived effective radius (Thomason et al., 1997), including the improvement to now incorporate HALOE data (Thomason, 2012), does provide additional constraints for the global variation in particle size. This is different to the Angstrom coefficient, which is based on only 2 wavelengths.*

*We have re-worded that paragraph slightly to instead read:*

```
"In the Pinatubo case, satellite measurements are able to provide
additional constraints for the particle size evolution, with
particle effective radius derived from the volume concentration and
surface area density SAGE-II extinction at multiple wavelengths
(Thomason et al., 1997; SPARC ASAP report, 2006). Hence for
Pinatubo, we also compare model-simulated effective radius to that
provided with the GloSSAC dataset, which underpins each climate
model's specified multi-wavelength aerosol optical properties in the
Pinatubo forcings in CMIP6 historical integrations."
```

Line 250: Mann et al (2019b) and (2020) are conference abstracts. Does ACP allow them as references?

*-->Yes, we followed ACP manuscript preparation guidelines for authors (https://www.atmospheric-chemistry-and-physics.net/for_authors/manuscript_preparation.html).*

Line 277: Larger injections produce a stronger upwelling (which push toward a longer S lifetime) and larger particle radii (which push toward shorter e-folding time). Do your result imply that the net effect is driven by the particle size, rather than the changes in upwelling?

*--> We agree with the reviewer that both effects are important, but although the result confirms the residence time occurs at the time of maximum effective radius, since the stronger upwelling is also linked to particle size changes, we do not feel the results support such a conclusive statement as the reviewer suggests.*

Line 295: Is there a published paper or report that documents the changes brought up by increasing the resolution? Maybe something was published when the model with higher resolution was released?

*--> Good point. The UK Met Office publishes documentation papers in GMD describing each successive Global Atmosphere configuration, with comparisons of climatologies of a range of different metrics and, for example, Walters et al. (2014) compare GA4, the physical model used in this study to the previous GA3 version.*

Line 306: How is stratospheric AOD calculated? Is aerosol extinction integrated above the tropopause or above a fixed altitude?

*--> Yes, aerosol extinctions are integrated for all the levels above the tropopause.*

Line 310: I'm confused by this. Are the authors referring to Fig.1, when they write that Pin20 best matches the satellite observed SO2 estimates? Pin20 is the one that compare the worst with HIRS.

*--> The sentence is referring to the 14-23 Tg range from Guo et al. (2004), we do not compare to SO2, but the emission amount of 20 Tg is in the upper-mid-range from the TOMS/TOVS satellite measurements. We are aware that Pin20 compares worst with HIRS derived SO4 burden, and this point is discussed clearly in the text.*

Line 313: I am not sure it is fair to say that Pin10 has the best agreement. All of them, including Pin10, overestimate the peak sAOD in the tropics. 18 months after the eruption Pin20 seems actually to do better. A similar statement requires a metric such as the globally averaged root mean square error. I generally find the qualitative comparison a weak point of this manuscript. It is very difficult to judge which simulation is performing best just by looking at the figures.

*--> As the reviewer notes the 10, 14 and 20 Tg simulations all overestimate the peak sAOD in the tropics, but Pin10 clearly has a lower high bias than the other scenarios, and that sense Pin10 has best agreement. We note in the text that the Pin20 run compares best in the Southern Hemisphere.*

Line 344: Cerro Hudson is at 45S, 12 km could be above the tropopause.

*--> Whilst we agree that the Hudson volcanic aerosol cloud reached the lowermost stratosphere, we just try to reiterate the point we make on line 344 that the measurements in the cited papers (e.g. Fig 1 of Pitts and Thomason, 1992) which demonstrate that Pinatubo was by far the dominant contributor to the stratospheric AOD, even when the Hudson cloud was at its maximum optical depth.*

Line 390: I think Pin20 should be included in Fig. 4. Even if high biased, it is interesting to see how the effective radius scales with injection burden.

*--> Done*

Line 392: I am confused by this sentence. Pin20 is not shown, and between Pin14 and Pin10 I don't see any clear difference. There is a need of some kind of metric, such as mean error. Judging from the current plot, both simulations seems to perform pretty poorly when compared to the CMIP6 dataset (if that is a valuable benchmark).

*--> We could have calculated a mean bias from the CMIP6 effective radius, but there is a sufficient variation among different effective radius datasets, with a substantial uncertainty, and for this reason we prefer not to calculate and evaluate metrics to just one dataset.*

Line 405: I am not sure where to look to see this. Please specify latitude and months of the part of the plot that you are commenting on.

--> As suggested by the reviewer, we now have additional panel for Pin20 in Figure 4, so

"At 25km, the model simulations are somewhat counter-intuitive. Initially, they show decrease in Reff, likely due to this central part of the volcanic cloud being younger (and smaller) particles formed as the oxidation of the volcanic SO2 triggers extensive new particle formation,"

is revised to instead read:

"The Reff increase is due to particle growth from coagulation and condensation, and the simulations illustrate much slower temporal increase in size at 25km than at 20km. The 25km level is in the central part of the volcanic cloud, particles there being younger (and smaller) as the oxidation of the volcanic SO2 continues to trigger extensive new particle formation. By contrast, at the 20km

level, particles there will almost exclusively have sedimented from the main cloud, and therefore at larger particle sizes. This explains why at 20km, below the altitude at which the volcanic plume detrains the SO2 (injection height range is 21-23 km) the effective radius shows a steady increase, as relatively larger particles sediment to these altitudes as the tropical volcanic aerosol reservoir progresses."

Line 598: Figure 13, not 12, right? Also, take out either "as" or "hence"

--> *Thanks for spotting this, we removed "hence".*

Figure 1: I think "blue line" should be "solid lines", otherwise I do not understand which lines I am supposed to look at.-

--> *This was an error in the caption, corrected to "Pin00 (aqua), Pin10 (blue), Pin14 (green), Pin20 (red)". Apologies for the confusion.*

Figure 2: I find this kind of graph (Fig 2, 5, 8, 7, etc) difficult to interpret. Next to the AOD, there should also be a panel with the absolute or relative difference between simulations and datasets. You could build a 3x3 table of graphs showing the difference between each of the three ensembles and each of the 3 datasets.

--> *Although it would be possible to construct the difference plot suggested, partly because of the differences between the observational datasets, and partly because it enables visual inspection of the patterns of variation in each dataset, we prefer to show side-by-side comparisons of the predicted metric rather than bias plots.*

Figure 3: just to be clear, the variability among ensemble members is the ensemble spread, right? Min to max values per each month.

--> *Yes. We consider the term "variability among ensemble members" more scientifically descriptive.*

Fig. 10: seeing the colors in panel d) is difficult, as lines become dense right where the warming happens. It would be better to make the lines light grey or change the color table to something with more diversity. Also, why not include the mean QBO also in the simulation graphs? Does the QBO change between the experiments with and without eruptions?

--> *We reduced the contour interval lines to 5 m/s, reduced line thickness and plotted contour lines with thicker black line to enhance the clarity.*

**Minor comments and typos.**

--> *Revised manuscript has been modified to add all the minor and technical corrections.*

**Additional comments added after the original upload of the above reply to reviewers**

*Shortly after uploading our replies to the reviewers (including AC2 above), and when finalising the revised manuscript, we discovered two subtle but important mistakes in the Python code used to generate the figures in the ACP-Discussions manuscript:*

1) *Figure 4: Typo in the code used to calculate effective radius: assigned the accumulation-soluble mode $H_2SO_4$ mmr to accumulation-insoluble $H_2SO_4$ mmr*

   *A subtle typo in the code used to calculate effective radius caused an error in the initial assignment of modal $H_2SO_4$ component mass mixing ratios. The typo caused the calculation of total particle volume (PVOL) to double-count the accumulation-soluble mode $H_2SO_4$ mass mixing ratios (mmr), the accumulation-insoluble $H_2SO_4$ mode mmr not used in the calculations as a consequence. Specifically, the excerpt of code:*

```
H2SO4_nucsol_mmr=nc_fid.variables['NUCLEATION_MODE__SOLUBLE__H2SO4_MMR'][:]
H2SO4_Aitsol_mmr=nc_fid.variables['AITKEN_MODE__SOLUBLE__H2SO4_MMR'][:]
H2SO4_accsol_mmr=nc_fid.variables['ACCUMULATION_MODE__SOL__H2SO4_MMR'][:]
H2SO4_corsol_mmr=nc_fid.variables['COARSE_MODE__SOLUBLE__H2SO4_MMR'][:]
H2SO4_accins_mmr=nc_fid.variables['ACCUMULATION_MODE__SOL__H2SO4_MMR'][:]
```

   *should have been*

```
H2SO4_nucsol_mmr=nc_fid.variables['NUCLEATION_MODE__SOLUBLE__H2SO4_MMR'][:]
H2SO4_Aitsol_mmr=nc_fid.variables['AITKEN_MODE__SOLUBLE__H2SO4_MMR'][:]
H2SO4_accsol_mmr=nc_fid.variables['ACCUMULATION_MODE__SOL__H2SO4_MMR'][:]
H2SO4_corsol_mmr=nc_fid.variables['COARSE_MODE__SOLUBLE__H2SO4_MMR'][:]
H2SO4_accins_mmr=nc_fid.variables['ACCUMULATION_MODE__INS__H2SO4_MMR'][:]
```

   *For the major volcanic aerosol cloud simulations analysed here, the majority of sulphuric acid mass is in that double-counted accumulation-soluble mode.*

   *Hence the typo caused the particle volume PVOL, used in the calculation of the model's effective radius (=3\*PVOL/SAREA) to be much higher.*

   *This caused the effective radius values shown in Figures 4 a), b), d), e) in the ACPD article to be much higher than their true values.*

2) *Figures 2d), 8d), 11d): Error in sAOD calculated for CMIP6 dataset (depth error).*

   *The stratospheric AOD values shown for the CMIP6 representations of the Pinatubo, El Chichon and Agung aerosol in Figures 2d, 8d, 11d of the ACPD article are factor 2 too high, due to an error in the depth used in the calculations.*

   *The depth error arises within our code to integrate to sAOD the altitude-resolved aerosol extinction dataset provided with the CMIP6 volcanic aerosol dataset.*

*When calculating the sum over vertical levels, the depth assigned when integrating the aerosol extinction to stratospheric Aerosol Optical Depth (sAOD) was set at 1.0km rather than 0.5km, calculated sAOD then a factor of 2 too high.*

*We apologise for both bugs. Whereas the error 1) was more subtle, and the Reff from the model being too high was not obvious, we should have realised that the CMIP6 sAOD values shown in those figures were a factor-2 too high. Even though there is no documentation paper for the pre-satellite part of the CMIP6 dataset (CMIP6-AER2D) sAOD, we should still have realised this error when preparing the manuscript. We are relieved to have found this error during the review process and in the revised manuscript, the Figures 2d, 8d and 11d show the correct sAOD$_{525}$ values.*

*The typo explained in 1) is now remedied, and the simulated Reff values shown in Figures 4c) to f) of the revised manuscript represent the model predictions correctly.*

*We also added the two extra panels requested by Reviewer 2 to additionally show the 20Tg simulation Reff field at 25km (Figure 4a) and 20km (Figure 4b).*

*Note that the Reff Figure in the Supplementary Material (Figure S6) has also been updated from the ACPD article to show the correct values. There only the 10Tg and 20Tg runs are shown to match the runs used in Dhomse et al. (ACP, 2014), enabling comparison with the corresponding figure in that paper.*

*With these changes to the simulated Reff values in Figure 4 and Figure S6, the Section 4.1 text analysing the Reff variations (lines 389-419 in the ACPD article) has been re-written to:*

"Next, we evaluate the meridional, vertical and temporal variations in effective radius (Reff) in the Pinatubo UM-UKCA datasets. The particle size variations in these interactive simulations of the Pinatubo cloud reflect the chemical and microphysical processes resolved by the chemistry-aerosol module, in association with the stratospheric circulation and dynamics occurring in the general circulation model. We analyse these model-predicted size variations also alongside those in the benchmark observation-based Reff dataset from CMIP6-GloSSAC, which applies the 3-□ size retrieval from the 453nm, 525nm and 1020nm aerosol extinction measurements from SAGE-II (Thomason et al., 1997a, 2018).

Figure 4 shows zonal mean Reff at 25km, within the altitude range of the volcanic SO$_2$ injection, and at 20km, underneath the main volcanic cloud, results shown from 3-member means from the 10, 14 and 20Tg SO$_2$ emission runs (Pin10, Pin14 and Pin20). For comparability with the equivalent Figure from Dhomse et al. (2014), we also show in the Supplementary Material (Figure S6) the updated comparison to the Bauman et al. (2003) Reff dataset, for the corresponding Pin10 and Pin20 runs. Overall, the model captures the general spatio-temporal progression in the Reff variations seen in

the GloSSAC dataset. However, whereas the 10Tg and 14Tg simulations agree best with the HIRS-2 sulphur-burden (Figure 1) and the GloSSAC sAOD and extinction (Figures 2 and 3), the magnitude of the Reff enhancement is best captured in the 20Tg run (Pin20). The comparisons suggest the low bias in simulated Reff seen in the previous UM-UKCA Pinatubo study (Dhomse et al., 2014) continues to be the case here. However, this low-bias in particle size/growth may simply be reflecting the required downward-adjustment of the Pinatubo $SO_2$ emission, a larger Reff enhancement in the 20Tg simulation clearly apparent. It is possible that the two-moment modal aerosol dynamics in GLOMAP-mode may affect its predicted Reff enhancement. However, the model requirement for reduced $SO_2$ emission is attributed to likely be due to a missing, or poorly resolved, model loss pathway, such as accommodation onto co-emitted volcanic ash. The sustained presence of ash within the Pinatubo cloud (e.g. Winker and Osborne, 1992) will likely have altered particle size and growth rates in the initial months after the eruption.

In the tropics, where Reff increases are largest, the timeseries of Reff is noticeably different in the core of the tropical reservoir (10°S to 10°N) to that in the edge regions (10°N-20°N and 10°S-20°S), at both 20km and 25km. The Reff increases in these edge regions occur when tropics to mid-latitude transport is strongest, in phase with the seasonal cycle of the Brewer-Dobson circulation, which tends to transport air towards the winter pole (Butchart, 2014). The Reff increases are due primarily to particle growth from coagulation and condensation, and the simulations also illustrate how the simulated Pinatubo cloud comprises much smaller particles at 25km than at 20km. The 25km level is in the central part of the Pinatubo cloud, particles there being younger (and smaller), because the oxidation of emitted volcanic $SO_2$ that occurs at that level, triggers extensive new particle formation in the initial months after the eruption (e.g. Dhomse et al., 2014). By contrast, at the 20km level, particles there will almost exclusively have sedimented from the main cloud, and therefore be larger. There is a slow but sustained increase in average particle size in the equatorial core of the tropical Pinatubo cloud, with the 20km level reaching peak Reff values only during mid-1992, in contrast to the peak S-burden and $sAOD_{550}$, which have already peaked at this time, being in decay phase since the start of 1992 (see Figures 1 and 2).

Whereas the simulated peak Reff enhancement occurs by mid-1992 in the tropics, the peak Reff in NH mid-latitudes occurs at the time of peak meridional transport, the Reff variation there reflecting the seasonal cycle of the Brewer-Dobson circulation, as also seen in the tropical reservoir edge region. The different timing of the volcanic Reff enhancement in the tropics and mid-latitudes is important when interpreting or interpolating the in-situ measurement record from the post-Pinatubo OPC soundings from Laramie (Deshler et al., 2003). Russell et al. (1996) show the Reff values derived from Mauna Loa ground-based remote sensing are substantially larger than those from the dust-sonde measurements at Laramie. The interactive Pinatubo simulation here confirm this expected meridional gradient in effective radius, with the chemical, dynamical and microphysical processes also causing a vertical gradient in the tropical to midlatitude Reff ratio. The current ISA-MIP activity (Timmreck et al., 2018) brings a potential opportunity to identify a consensus among interactive stratospheric aerosol models for the expected broad-scale spatio-temporal variations in uncertain volcanic aerosol metrics such as effective radius.

*With the 1km-depth error in the integration of the CMIP6 aerosol extinction, and the subsequent correction to the sAOD shown for CMIP6-GloSSAC/CMIP6-AER2D in Figures 2d, 8d and 11d, there have also been some minor changes to interpret the evaluation of the UM-UKCA volcanic simulations. The revisions here are only minor changes in emphasis re: the comparisons to GloSSAC sAOD, and since the text is mainly analysing the sAOD variation, these changes are only minor.*

*The text changes for this CMIP6 sAOD correction are the lines 371-415, 562-583 and 623-631 of the revised (lines 446-491, 672-694 and 735-755 in a tracked change version) manuscript, corresponding to lines 316-357, 484-504 and 542-563 of the ACP Discussions article*

---

## Author Comment (AC3) · 26 Aug 2020

Dear Reviewer, while finalising the revised manuscript, we discovered two subtle but important mistakes in the Python code used to generate the figures in the ACP-Discussions manuscript:

1. Figure 4: Typo in the code used to calculate effective radius: assigned the accumulation-soluble mode H2SO4 mmr to accumulation-insoluble H2SO4 mmr.

2. Figures 2d), 8d), 11d): Error in sAOD calculated for CMIP6 dataset (depth error).

The stratospheric AOD values from CMIP6 dataset shown Figures 2d, 8d, 11d of the ACPD article are factor 2 too high, due to an error in the depth used in the calculations.

Detailed explanation about those errors in provided at the end of modified Authors response document.

---

## Author Comment (AC4) · 26 Aug 2020

Dear Reviewer, while finalising the revised manuscript, we discovered two subtle but important mistakes in the Python code used to generate the figures in the ACP-Discussions manuscript:

1.  Figure 4: Typo in the code used to calculate effective radius: assigned the accumulation-soluble mode $H_2SO_4$ mmr to accumulation-insoluble $H_2SO_4$ mmr.

2.  Figures 2d), 8d), 11d): Error in sAOD calculated for CMIP6 dataset (depth error).

The stratospheric AOD values from CMIP6 dataset shown Figures 2d, 8d, 11d of the ACPD article are factor 2 too high, due to an error in the depth used in the calculations.

Detailed explanation about those errors in provided at the end of modified Authors response document.

---

## Author Response (AR2)

Topical Editor Technical corrections:

- Reff should be written with "eff" as subscript throughout the manuscript.
Yes – it's corrected -- done.

P4, L90: Typo in methodologies (third "o" is missing)
Yes – it's corrected -- done.

P10, L292: add "a" -> with a number of models.....
Yes – it's corrected -- done.

P10, L298: add "a" -> generate a peak loading ....
Yes – it's corrected -- done.

P10, L309: remove space between Pin20 and closing parentheses.
Yes – it's corrected -- done.

P11, L349: space between circulation and opening parentheses missing.
Yes – it's corrected -- done.

P11, L355: Abbreviation OPC has not been introduced.
Yes – it's corrected -- on line 341 changed "OPC" → "optical particle counter (OPC)" (341 is 1st use).
Note we also made a change after realising our statements about the HIRS data on lines 355-359 were incorrect – we changed "is inconsistent with" to "is partially consistent with" (lines 354-355) and also changed "may be inaccurate" to "may well be accurate".  We're sorry for our mistake there.

P12, L361: add "in the" so that it reads "...are shown in the Supplementary Figures......
Yes – it's corrected also modified as ", these are shown

P12, L366: Abbreviation NH has not been introduced yet.

P13, L397: Abbreviation SH has not been introduced yet.,

Yes – these set to "Northern Hemisphere (NH)" and "Southern Hemisphere (SH)" -- done.

P14, L431: add "to" -> leading to faster.....

Yes – it's corrected -- done.

P16, L512: add "to" -> relative to the.....

Yes – it's corrected – done.

P16, L513: "the" twice, thus one is obsolete.

Yes – it's corrected -- done.

P16, L515: Check sentence. Do you mean the modelled Pinatubo forcings?
Yes – it's corrected -- to "the Pin20 simulation also over-predicts the magnitude of the Pinatubo forcing compared to the ERBE anomaly" – done.

P16, L518: Add space between number and unit.

Yes – it's corrected -- done.

P16, L530: add "a" -> might be a much weaker..... or write plural "much weaker signals".

Yes – it's corrected – from "might be much weaker" to "might be poorly sampled" -- done.

P17, L545: add "that the" -> predicts that the Pinatubo aerosol.......

Yes – it's corrected -- done.

P17, L546: as in? I would suggest to write either as found in or as seen in.

Yes – it's corrected -- done.

P17, L561: stratospheric -> stratosphere

Yes – it's corrected -- done.

P17, L561: reanalyses or re-analyses, it should be written one way consistently throughout the manuscript.

Yes – it's corrected -- done.

P17, L562: Figure 8 is referenced before Figure 7. It seems that the reference to Figure 7 is missing at all.
Yes – we have corrected that sentence to say:
"Figure 7 shows ERA-interim temperature anomalies compared to the model's volcanic warming, i.e. the temperature difference between the sensitivity (Pin10, Pin14 and Pin20) and control (Pin00) simulations".

P18, L576: add "the" -> via the middle branch.....

Yes – it's corrected -- done.

P18, L576: at around -> better to write to around or up to around.

Yes – it's corrected -- done.

P18, L585: better to write "at extinction of"?

Yes – it's corrected – done.

P18, L589: removal of what? Please add.

Yes – it's corrected – done.

P19, L624: add "the" -> suggest that the tropical ......

Yes – it's corrected -- done.

P19, L634: use plural -> different responses between two CMIP assessments.

Yes – it's corrected – we re-worded the sentence to say:
"Hence, the historical climate integrations performed using these two forcing datasets will have simulated substantially different Agung surface cooling between the two CMIP assessments."

P19, L635: Sato data -> The Sato data

Yes – it's corrected -- done.

P20, L636: add "a" -> within a couple of months

Yes – it's corrected – we preferred to re-word to "just a few months after the eruption".

P20, L636:Thenafter -> Thereafter

Yes – it's corrected – done.

P20, L642: add "the" -> to the CMIP6 datasets

Yes – it's corrected – done.

P20, L642: add "a" -> predict a secondary

Yes – it's corrected -- done.

P20, L651: add "that" -> It is notable that those....

Yes – it's corrected -- done.

P21, L673: Add "the" -> Although the lidar

Yes – it's corrected -- done.

P21, L673: add "the" -> ...the Agung aersol

Yes – it's corrected -- done.

P21,L676: Rewrite sentence. Either "modelled tropics to NH midlatitude transport" or " weak transport from tropics to NH midlatitudes in the model"

Yes – it's corrected – and we re-worded slightly to improve the wording.

P21, L677: add "the" and "S" in capital letter -> the Supplementary Figure

Yes – it's corrected -- done.

P21, L678: add "the" -> the Lexington lidar
Yes – it's corrected -- done.

P21, L684: add "an" -> estimated an about 1.5.K

Yes – it's corrected – and we re-worded that sentence to say:
"Using radiosonde data, Free and Lanzante (2009) attributed around 1.5 K warming to the Mt. Agung volcanic aerosol"

P21, L685: add "the" -> due to the westerly QBO

Yes – it's corrected -- done.

P21, L699: 1991-3? Do you mean 1991-1993?

Yes – it's corrected -- done.

P22, L704: add "a" -> it is a common feature

Yes – we re-worded the sentence to:

*"The model ensembles compare very well to mid-visible and near-infra-red aerosol extinction from SAGE-II measurements, although the model has sAOD high bias in the tropics, a common feature seen in interactive stratospheric aerosol models* (Niemeier et al., 2009; Mills et al., 2016; Sukhodolov et al., 2018)". N.b. we changed that to cite the 2009 Niemeier study :

Niemeier, U., Timmreck, C., Graf, H.-F., Kinne, S., Rast, S. and Self, S.
"Initial fate of fine ash and sulfur from large volcanic eruptions"
Atmos. Chem. Phys., 9, 9043–9057, 2009.

P22, L709: rage -> range

Yes – it's corrected -- done.

P22, L728: Sato data -> The Sato data

Yes – it's corrected -- done.

P22, L733: suggests that -> suggest a

We re-worded that sentence to read:

"Comparison with the lidar measurements from Lexington suggests that the UM-UKCA simulated vertical extent of the Agung cloud in the NH mid-latitudes is in best agreement for run Agu09."

Figure 7, 10, 13: Colour bar label cut and in Figure 13 also the x-axes labels are cut.

Yes – it's corrected -- done.

Supplement, P2, L24: space between numbers and units missing.

Yes – it's corrected – done.  We also realised we missed out an important part of the methodology to explain the 1-year time offset we applied re: applying the extinction-to-backscatter ratio values from Pinatubo (from Jaeger and Deshler, 2003) to the Agung cloud.

So we also added the sentence "Since the Northern Hemisphere Agung cloud had much lower optical thickness than the Pinatubo cloud, we also included a 1-year time offset, with the 1993 and 1994 extinction-to-backscatter ratio values used for Lexington data in 1964 and 1965 respectively."

Supplement, P3, Fig S1 caption: One monthly mean obsolete? Shouldn't it read "Monthly mean tropical stratospheric zonal mean wind?

Yes – it's corrected -- done.

Supplement, P3, Fig S1 caption: in Pin 10.... -> of the Pin 10....

Yes – it's corrected -- done.

Supplement, P4, Fig S2 caption: in -> of the

Yes – it's corrected -- done.